

# Climate-induced hydrologic change in the source region of the Yellow River: a new assessment including varying permafrost

Pan Wu[1], Sihai Liang[1], Xu-Sheng Wang[1], Yuqing Feng[1], Jeffrey M. McKenzie

[1]School of Water Resources and Environment, China University of Geosciences (Beijing), Beijing, 100083, P. R. China.

[2]Earth and Planetary Sciences, McGill University, Montreal, Quebec, H3A 0E8

*Correspondence to*: Sihai Liang (liangsh@cugb.edu.cn)

**Abstract.** The source region of the Yellow River (SRYR) provides 35% of the rivers annual discharge but is very sensitive to the climate change. The change in discharge from the SRYR has been attributed to both climatic and anthropogenic forces, and previous estimates of the impact of human activities on the change in discharge have been higher than 50% of the total change. Considering the very low
population density and limited land use change, this result is potentially inconsistent. Our study modifies the traditional Budyko separating approach to identify and quantify the climatic causes in discharge changes. Application of this new approach to the SRYR now highlights the role of the degrading permafrost, based on long-term observation data of the maximum frozen depth (MFD). Our results show that over the past half-century, the change in discharge in the SRYR was primarily
controlled by climate change rather than local human activities. Increasing air temperature is generally a negative force on discharge whereas it also causes permafrost to degrade - a positive factor on discharge generation. Such conflicting effects enhance the uncertainty in assessments of the hydrological response to climate change in the SRYR.

## 1. Introduction

In the past five decades, climate has changed dramatically at the global scale (IPCC,2007), resulting in long-term impacts on global river systems (Abdul Aziz and Burn, 2006; Birsan et al., 2005; Chen et al., 2014; Chiew et al., 2009). The response of hydrological basins in the Tibetan Plateau due to climate change is a major concern for the management of water resources, particularly in China. The water balance of headwater basins in the Tibetan Plateau is influenced by climate change at different levels
(Immerzeel et al., 2010; Liang et al., 2013; Su et al., 2016). Originating from the Tibetan Plateau, the Yellow River is the second longest river in China. The river's streamflow intensity has undergone large changes in the past 50 years, primarily observed by a sharp decreased in many sub-basins during the 1990s (Liu and Zheng, 2004) . Due to the dry climate and heavy water demands, people in the Yellow River basin are facing serious water shortages (Yang et al., 2004).





The source region of the Yellow River (SRYR) is a large basin above the Tangnaihai hydrological station (Figure 1) which contributes about 35% of the total annual streamflow, but has an area that is only 16% (121,972 km$^2$) of the total Yellow River basin (Lan et al., 2010). In the 1990s, streamflow from the SRYR significantly decreased with increasing cases of flow cutoff (Yang et al., 2004; Zhang et al., 2008; Zheng et al., 2007). As shown in Table 1, the causes of the decreased flow within the SRYR and other sub-basins of the Yellow River have been extensively investigated in the literature. However, there are disagreements as to the causes of the observed decreases in streamflow. Many researchers suggested that climate change (manifested as decreased precipitation and increased evapotranspiration) is the primary cause of the decrease in streamflow (Cuo et al., 2013; Liang et al., 2010; Meng et al., 2016; Tang et al., 2013). Other researches proposed that the decrease in discharge was mainly caused by changes in catchment properties (Zheng et al., 2009, Zhao et al., 2009). For example, Zheng (2009) found that the decrease in streamflow in the 1990s was a result of climate change and land use change but highlighted that the land use change was more important and that it contributed to about 70% of the total change in streamflow. Zhao (2009) suggested that 65% of the streamflow change in the upper catchment of the Jimai hydrological station could be attributed to the effects of human activities. This conclusion is potentially at odds with the very low population density in the area above the Tangnaihai hydrological station (about 6 people/km$^2$, 2003 census data) and in the area above the Huangheyan station (0.34 people/km$^2$) (Liang et al., 2010). From 1990 to 2000, the change in land use in the SRYR was generally less than 5%, though a few sites exhibited 5-15% of the change (Wang et al., 2010). Therefore, it is not easy to attribute the negligible impact of human activities and land use changes to more than 50% of the change in the discharge of the SRYR.

One reason for the divergent (and potentially problematic) findings from previous analysis on the climate elasticity of the streamflow in the SRYR (Zheng et al., 2009, Zhao et al., 2009) may be the misattribution of the change in catchment properties. The concept of climate elasticity of discharge was widely used to separate the change in streamflow into different components, including climate changes, landcover changes and human activities that induced by changes in climatic forces and catchment properties (Arora, 2002; Fu et al., 2007; Zheng et al., 2009; Yang and Yang; 2011). The sensitivity method (Roderick and Farquhar, 2011) and decomposition method (Wang and Hejazi 2011) were also developed with the process like the climate elasticity analysis method. Previously the effect of change in catchment properties was usually attributed to human activities or land use change (Ma et al., 2008, Zheng et al., 2009, Zhao et al., 2009). This attribution would lead to a bias in the findings because the catchment properties could also depend on the climate condition (Yang et al., 2007; Williams et al., 2012; Jiang et al. 2015). Thus, the change in catchment properties in the SRYR should be re-examined.

The other poorly known factor in the SRYR is the hydrological impacts of the changes in the permafrost distribution. Permafrost is defined as a subsurface region in which the temperature is perennially below 0 ˚C. The SRYR is widely underlain by permafrost, particularly in the area above the



Jimai station (Figure 1; Zhou et al., 2000) where the altitude is relatively high. In the recent 50 years, the air temperature in the SRYR increased at an average rate between 0.31°C /10a and 0.35°C /10a (Cuo et al., 2013; Hu et al., 2011). Due to this warming trend, permafrost has warmed and degrading, and the depth of the annual active layer has increased in the upper catchment above Jimai station (Luo

et al., 2014a; Luo et al., 2014b). Permafrost degradation will increase the depth and length of subsurface flow paths and the lag-times of subsurface water flow from infiltration or permafrost thawing to surface water discharge (Frampton et al., 2011, Kurylyk et al., 2014). Degrading permafrost could also accelerate the flow recession process and increase local hydrological circulation (Lyon and Destouni, 2010; Lyon et al., 2009; Cuo et al., 2015), change aquifer permeability, and increased base

flow (Walvoord and Striegl, 2007; Bense et al., 2012; Evans et al. 2015). These relationships have not been previously examined in the SRYR even though many researches have focused on the dramatic decreases in streamflow in the 1990s.

The objective of this study is to develop a new assessment of the causes of discharge changes in

different sub-basins of the SRYR. A specific catchment parameter, based on the Budyko framework, is applied to represent the properties of the drainage basins. Correlation analysis with multiple models is performed to reveal the relationships between the time-varying catchment parameter and potential drivers of hydrologic change, such as the climatic forcing and permafrost degradation.

## 2 Study area and data processing

### 2.1 Catchments and sub-basins

The Yellow River originates from the Tibetan plateau, flows across northern China from west to east, and eventually empties into the Bohai Sea. The source region of the Yellow River (SRYR) is in the northeastern Tibetan Plateau where there is extensive permafrost (Figure 1). The Tangnaihai hydrological station is at the outlet of the SRYR, and receives the discharge from the 121,972 km$^2$

catchment area. The SRYR contributes about 35% of the total annual river flow (Lan et al., 2010) so that it is an essential site for the Chinese water resources. Two major lakes, Gyaring and Ngoring, are located within the highest sub-basin controlled by the Jimai hydrological station. Another hydrological station, Maqu, monitors the middle reaches between Jimai and Tangnaihai.

The landscape of the study area is characterized by permafrost that has undergone significant degradation since the 1980s (Jin et al., 2010). The distribution of the permafrost in the 1980s is reported in the Map of Snow, Ice, and Frozen Ground in China (1988) published by the Environmental and Ecological Science Data Center for West China, National Natural Science Foundation of China (http://westdc.westgis.ac.cn). For this map, there are three permafrost classifications: predominantly

continuous permafrost, isolated permafrost and alpine permafrost. This classification scheme is different than that of the International Permafrost Association (IPA) (Cheng and Wu, 2007; Ren et al.,





2012). In Figure 1b, the predominantly continuous permafrost and the isolated permafrost are combined into the continuous permafrost for simplification. Due to the water resource significance and unique landscape, the SRYR provides an ideal location to observe the hydrological effects of degrading permafrost with climate change.

In this study, the SRYR area is divided into three sub-basins from the headwaters to the SRYR outflow: the Jimai (JM) basin, the Jimai-Maqu (MQ) basin, and the Maqu-Tangnaihai (TNH) basin. Historical observation data are collected and re-organized for the three sub-basins. A significant difference between the sub-basins is the coverage area of the permafrost. As shown in Figure 1b, almost all the
JM (80%) is covered by the continuous permafrost whereas in the MQ only 23% of the basin area has is occupied by the continuous permafrost and alpine permafrost. In the TNH, 46% of the basin area is occupied by the permafrost.

## 2.2 Data processing

Annual discharge data since 1961 at the Jimai, Maqu, Tangnaihai stations were obtained from the
Yellow River Conservancy Commission (YRCC) and Loess Plateau Data Center (http://loess.geodata.cn). Daily meteorological data between 1961 and 2013 at 18 meteorological stations (Figure 1b) within the SRYR or in the surrounding areas were collected from the China Meteorological Administration (CMA), including precipitation, air temperature, wind speed at 10 m height, relative humidity and sunshine duration. The potential evapotranspiration was calculated with
the FAO Penman-Monteith equation  (Allen et al., 1998). These data were applied to derive the annual precipitation ($P$, mm), annual precipitation intensity ($I$, mm/d), annual mean temperature ($T$, °C) and annual potential evapotranspiration ($E_0$, mm) from 1961 to 2013 at the meteorological stations.

The annual precipitation intensity, $I$, was calculated at the meteorological stations, and defined as the
sum of daily precipitation divide by the number of days with rainfall for every year. Then, gridded data of $P$, $E_0$, $T$ and $I$ were created using the inverse distance weighted technique (IDW), with the resolution of 0.01° × 0.01°. Based on the gridded data, the mean values of $P$, $E_0$, $T$ and $I$ in the three sub-basins were calculated for each year from 1961-2013.

We do not correct the meteorological data for elevation effects because the properties of the elevation effects are poorly known in the SRYR. However, it is expected that the elevation-dependent distributions of these factors are mostly included within the IDW results. In general, the elevation of ground surface in the study area increases gradually from the east to the west and creates a climate gradient parallel to the altitude gradient that is exhibited in the differences in the observed
meteorological data (Figure 1b). For the annual precipitation, we found that these IDW data at the catchment scale agreed with TRMM data (Tropical Rainfall Measurement Mission) 3B42 data (Tong et





al., 2014) which show a great performance in streamflow simulation in upper Yellow and Yangtze River basins on the Tibetan Plateau (Hao et al., 2014). The IDW interpolation is adopted in this study to fully capture the collected $P$, $E_0$, $T$ and $I$ data at the weather stations.

Daily observations of frozen ground at 11 meteorological stations (Figure 1b) was collected by CMA. The daily frozen depth of the active layer was identified every month and then used to estimate the mean value of the maximum frozen depth (MFD) for each year of the study period. The MFD value obtained from the maximum monthly mean frozen depth for each year was used to indicate the degradation of permafrost, and represents the mean frozen depth of the active layer for a given year.

Typical variation patterns of the mean frozen depth are shown in Figure 2. The mean frozen depth exhibits a decreasing trend at most of the observation sites, especially in the period after 1980. The MFD data of the observation stations in and around each sub-basin show the same tend (Figure 2). In considering of the different record lengths of MFD in different observation stations, a station with the longest record is selected to account for the representative MFD data in each sub-basin. Therefore, M5,

M10, and M14 are selected for JM, MQ and TNH, respectively, as that shown in Figure 1b. The station M14 is not located in the MQ sub-basin but it exhibits the same variation trend of MFD with that observed at stations in the MQ basin (M12, M13 and M16) with shorter records. However, M17 shows little variation compared to the other sites because it is located at alpine permafrost and therefore is not used in this study.

Figure 3 shows that discharge change can be divided into three periods: pre-change period from 1961-1990, low-flow period from 1991-2002 and recent period from 2003-2013. The two break points (1990, 2002) are also adopted by Tang et al. (2013). The baseline period is same as the climate pre-change period defined by IPCC (IPCC, 2007).

**3 Methods**

To assess the impact of degrading permafrost on the hydrology of the sub-basins, we use a new analysis approach compared to previous studies from the SRYR. First, the Budyko framework is applied to investigate the shift of mean annual water balance. Second, the change in the discharge is directly partitioned into climatic and anthropic components, using three existing methods in previous

studies based on the Budyko framework. A modified method is introduced using a covariate analysis to further partition the observed changes in discharge into two components: correlation with changes in $P(I)$, $E_0$ and $T$, and unknow forcing. Finally, the relationship between the unknow forcing change in the discharge and the degrading permafrost was investigated.





### 3.1 Budyko framework

Evapotranspiration in a catchment can be calculated from the water balance equation as follow:

$$E = P - Q - \Delta S \tag{1}$$

where the annual evapotranspiration, $E$, the annual precipitation, $P$, and the annual discharge, $Q$, are calculated as the average flux to and from the basin, whereas $\Delta S$ is the increment of storage during an accounting period. In a long period (usually more than 10 years), the change in storage ($\Delta S$) is negligible for the mean annual water balance so that $E = P - Q$ can be applied.

Budyko (1974) demonstrated that the mean annual evapotranspiration ratio ($E/P$) is mainly controlled by the mean aridity index ($= E_0/P$ where $E_0$ is the annual potential evapotranspiration). This hypothesis is the underlying principle of the Budyko framework in building a straightforward estimation formula for $E/P$ versus $E_0/P$. Non-parametric and one-parameter formulas have been proposed in the literature (Table 2). Non-parametric formulas were proposed before 1980 to represent a deterministic regression

curve for the relationship between $E/P$ and $E_0/P$ from a global dataset. One-parameter formulas were developed after 1980 to introduce catchment specific parameters ($w$, $n$ and $\alpha$ in Table 2) so that a group of regression curves for the relationship between $E/P$ and $E_0/P$ were formed for different types of catchments. The one-parameter formulas were more practical in comparison to the non-parametric formulas where the specific catchment parameter can represent the variable catchment properties

(Xiong and Guo, 2012). Accordingly, the long-term mean evapotranspiration ratio can be calculated as follows:

$$\frac{E}{P} = F\left(\frac{E_0}{P}, \ w\right) \tag{2}$$

where $w$ is the specific catchment parameter, $F()$ denotes an empirical formula. The long-term mean discharge $Q$ can then be calculated as:

$$Q = P - E = P\left(1 - F\left(\frac{E_0}{P}, w\right)\right) = f(P, E_0, w) \tag{3}$$

here $f()$ is the function derived from F(). The parameter, $w$, can be inversely determined with observation data of $Q$, $E_0$, and $P$, for a long period using the mean annual water balance, or for year-by-year patterns using a moving average method (Jiang et al., 2015).

### 3.2 Existing separating methods for climatic and anthropic causes of the change in discharge

In general, the change in discharge ($\Delta Q$) is attributed to the impacts of both climate change and human activities. Equation (3) shows the general response of discharge on the climate so that it can indicate the natural climatic change in discharge ($\Delta Q_c$). Thus, the difference between the actual change ($\Delta Q$) and the climatic change ($\Delta Q_c$) can be attributed to the anthropic cause, which herein is denoted as $\Delta Q_h$.




Three existing methods have been developed in the literature to partition $\Delta Q$ into $\Delta Q_c$ and $\Delta Q_h$, using different definitions. However, as compared by Wang and Hejazi (2011), these separation methods obtain similar results when they were applied for the same catchments.

### 3.2.1 Climate elasticity method

The concept of climate elasticity was proposed by Schaake (1990) for one variable (Precipitation). This method has been subsequently modified to considering two variables, precipitation and evaporation (Arora, 2002; Dooge et al., 1999; Kuhnel et al., 1991), and is described as:

$$\Delta Q_c = \frac{\partial f}{\partial P}\Delta P + \frac{\partial f}{\partial E_0}\Delta E_0 = f'_P \Delta P + f'_{E_0} \Delta E_0 \tag{4}$$

where $f'_P$ and $f'_{E_0}$ are the partial derivatives of $f()$ in Eq. (3) by treating the catchment specific

parameter ($w$) as a constant. The $w$ value must be determined for a period in the natural state and then regarded as a constant. The anthropic change ($\Delta Q_h$) can then be obtained by:

$$\Delta Q_h = \Delta Q - \Delta Q_c \tag{5}$$

### 3.2.2 Sensitivity method

The sensitivity method differs from the climate elasticity method as the catchment specific parameters

are variable (Roderick and Farquhar, 2011):

$$\frac{dQ}{Q} = \left[\frac{P}{Q}f'_P\right]\frac{dP}{P} + \left[\frac{E_0}{Q}f'_{E_0}\right]\frac{dE_0}{E_0} + \left[\frac{w}{Q}f'_w\right]\frac{dw}{w} \tag{6}$$

where the terms in the square brackets are called the sensitivity coefficients, $f'_P$, $f'_{E_0}$ and $f'_w$, and they directly represent the sensitivities of the discharge with respect to the changes in $P$, $E_0$ and $w$, respectively. Accordingly, the change in discharge can be estimated as:

$$\Delta Q = f'_P \Delta P + f'_{E_0} \Delta E_0 + f'_w \Delta w \tag{7}$$

To eliminate discretization errors, $f'_P$, $f'_{E_0}$ and $f'_w$ in Eq. 7 are substituted with $f^*_P$, $f^*_{E_0}$ and $f^*_w$, respectively (Jiang et al., 2015), as follow:

$$f^*_P = \frac{1}{2}\left[f'_P(P,E_0,w) + f'_P(P+\Delta P, E_0+\Delta E_0, w+\Delta w)\right] \tag{8}$$

$$f^*_{E_0} = \frac{1}{2}\left[f'_{E_0}(P,E_0,w) + f'_{E_0}(P+\Delta P, E_0+\Delta E_0, w+\Delta w)\right] \tag{9}$$

$$f^*_w = \frac{1}{2}\left[f'_w(P,E_0,w) + f'_w(P+\Delta P, E_0+\Delta E_0, w+\Delta w)\right] \tag{10}$$

The climatic and anthropic changes in discharge are considered as





$$\Delta Q_c = f_P^* \Delta P + f_{E_0}^* \Delta E_0 , \ \Delta Q_h = \Delta Q_w = f_w^* \Delta w \tag{11}$$

where $\Delta Q_w$ is applied in this study to denote the change in discharge induced by the change in the specific catchment parameter. This method estimates the variable $w$ values and assumes that $\Delta w$ is caused by the human activities such as the change in land use.

### 3.2.3 Decomposition method

For the decomposition method (Wang and Hejazi, 2011), different parts of the change in discharge are estimated from:

$$\Delta Q_c = f(P + \Delta P, E_0 + \Delta E_0, w) - f(P, E_0, w) \tag{12}$$

$$\Delta Q_h = f(P, E_0, w + \Delta w) - f(P, E_0, w) = \Delta Q - \Delta Q_c \tag{13}$$

From Eq. 13 we see that this method also attributes $\Delta w$ to human activities.

### 3.3 Identifying the climatic impact on the catchment specific parameter

In the existing approaches, the change in the catchment specific parameter and the induced change in
the discharge are attributed to human activities, *e.g.*, change in land use. However, this is not true because the catchment specific parameter can also be influenced by the natural processes following climate change (Yang et al., 2007; Williams et al., 2012; Jiang et al. 2015). Thus, it is necessary to further separate $\Delta w$ into climatic change, $\Delta w_c$, and anthropic change, $\Delta w_h$, as:

$$\Delta w = \Delta w_c + \Delta w_h \tag{14}$$

where $\Delta w_c$ is accounts for estimating the climatic change in the discharge.

It is difficult to directly separate $\Delta w_c$ and $\Delta w_h$ from the $w$ values year-by-year during a given period because the relationship between $w$ and the catchment conditions is not well known in the literature. In this study, we apply an indirect method which is similar to the approach in Jiang et al. (2015). The $w$
value is assumed to be linearly correlated with some explanatory variables as follow

$$w = \beta_0 + \beta_1 \frac{Pre}{Pre_m} + \beta_2 \frac{E_0}{E_m} + \beta_3 \frac{T}{T_m} + X \tag{15}$$

where $\beta_0$, $\beta_1$, $\beta_2$ and $\beta_3$ are coefficients, $T$ is the annual mean air temperature, $Pre$ represents the characteristics of precipitation (two forms was considered: precipitation, $P$, or precipitation intensity, $I$) and $E_0$ is the annual potential evapotranspiration.



Each of the variables is normalized by the absolute mean annual value, indicated by subscript $m$ ($Pre_m$, $E_m$ and $T_m$). $T_m$ values are 1.48°C, 1.35°C and 1.27°C for JM, MQ and TNH, respectively. The annual precipitation intensity was extracted for the sub-basins in data-processing (Sect. 2.2). Due to the very low population density (2003 census data; Liang et al., 2010) and a slight land use change (Wang et al.,

2010) in the SRYR, human activities were not directly considered in this study. Thus, $X$ is taken as a residual error from the linear regression with $\beta_0$, $\beta_1$, $\beta_2$ and $\beta_3$.

Similar to using Eq. (15), Jiang et al. (2015) proposed a general linear function to represent the dependency of $w$ on a number of possible explanatory variables and determined the $\beta_i$ ($i=0, 1, 2…$)

values through the covariate analysis and the Akaike's information criterion (AIC) for multiple models. Three functional forms (i.e., identity, exponential, and logarithmic) may be used for each variable, however, only the identity form was used in this study for simplification. The $\beta_i$ ($i=0, 1, 2…$) values were determined through stepwise regression analysis. In each step of stepwise regression analysis, a variable is considered for addition to, or subtraction from, the set of explanatory variables based on a

sequence of F-tests, and the variance inflation factor (VIF) was used to diagnose the collinearity. The selected model has the highest R-squared value and contains variables with VIF< 10. A high VIF value indicates the presence of multicollinearity, a VIF value of 10 is often adopted to diagnose the multicollinearity. In practice, VIF < 10 indicates ignorable multicollinearity among these variables (Neter et al., 1989; Hair et al., 1995; Jolliffe, 2002; Jou and Cho, 2014).

According to Eq. (15), the climatic change in the specific catchment parameter, $\Delta w_c$, can be estimated as

$$\Delta w_c = \beta_1 \Delta \overline{Pre} + \beta_2 \Delta \overline{E}_0 + \beta_3 \Delta \overline{T} \tag{16}$$

where $\overline{Pre}$ ($=P/P_m$ or $I/I_m$), $\overline{E}_0$ ($=E_0/E_m$) and $\overline{T}$ ($=T/T_m$) are the normalized value of $P$ (or $I$), $E_0$ and $T$,

respectively. However, Eq. (16) misses the impact of the change in permafrost and would be not sufficient to represent all climatic forces on the catchment parameter in the study area. Thus, a modified formula is proposed in this study to check the impact of permafrost change:

$$\Delta w_c = \beta_1 \Delta \overline{Pre} + \beta_2 \Delta \overline{E}_0 + \beta_3 \Delta \overline{T} + \beta_4 \Delta \overline{D} \tag{17}$$

where $\overline{D}$ and $\beta_4$ are the normalized MFD used to indicate the status of the permafrost and the relevant

coefficient, respectively. The term $\beta_4 \overline{D}$ should be included in $X$ in Eq. (15) and can be identified from the correlation between $X$ and $\overline{D}$.

In a different way from the existing decomposition methods, we define:





$$\Delta Q_w = f\left(P, E_0, w + \Delta w\right) - f\left(P, E_0, w\right) \tag{18}$$

$$\Delta Q_{c1} = f\left(P + \Delta P, E + \Delta E_0, w\right) - f\left(P, E_0, w\right) \tag{19}$$

$$\Delta Q_{c2} = \frac{\Delta w_c}{\Delta w} \Delta Q_w \tag{20}$$

and take the climatic change in discharge as the summation of $\Delta Q_{c1}$ and $\Delta Q_{c2}$:

$$\Delta Q_c = \Delta Q_{c1} + \Delta Q_{c2} \tag{21}$$

where $\Delta Q_{c1}$ equals to $\Delta Q_c$ in Eq. (13), which depends on the changes in $P$ and $E_0$; and $\Delta Q_{c2}$ is the additional climatic change in the discharge induced by $\Delta w_c$. The anthropic change in discharge can then be estimated as $\Delta Q_h = \Delta Q - \Delta Q_c$. The comparison between results of Eq. (16) and Eq. (17) indicates the impacts of the permafrost degradation on runoff in the SRYR, as presented in Sect. 4.2 and 4.3.

### 3.4 Selection of the Budyko formula

Among the one-parameter formulas (Table 2), the Fu's equation (Fu, 1981; Zhang et al., 2004) yields the first analytical solution of the Budyko hypothesis. The equation proposed by Zhang et al. (2001) is an empirical solution which does not agree with the wet boundary condition when $w=2$ (Yang et al., 2008). Yang et al. (2008) presented a new analytical derivation of Choudhury's (1999) solution and obtained a very high ($R^2=0.999$) linear correlation between the catchment specific parameter in the Fu's equation and that in the Choudhury's equation. Similar results were obtained by other researches based on Fu's equation and Choudhury's equation (Yang et al., 2008, Wang and Hejazi, 2011, Jiang et al., 2015). To avoid repetition, only the Fu's equation is applied in this study.

Substituting the Fu's equation into Eq. (3), we get:

$$Q = f\left(P, E_0, w\right) = \left(P^w + E_0{}^w\right)^{\frac{1}{w}} - E_0 \tag{22}$$

Accordingly, $f_P'$, $f_{E_0}'$ and $f_w'$ are given by:

$$f_P' = P^{w-1}\left(P^w + E_0{}^w\right)^{\frac{1-w}{w}} \tag{23}$$

$$f_{E_0}' = E_0{}^{w-1}\left(P^w + E_0{}^w\right)^{\frac{1-w}{w}} - 1 \tag{24}$$

$$f_w' = \frac{\left(P^w + E_0{}^w\right)^{\frac{1}{w}}}{w}\left[\frac{P^w lnP + E_0{}^w lnE_0}{P^w + E_0{}^w} - \frac{ln\left(P^w + E_0{}^w\right)}{w}\right] \tag{25}$$





To inversely obtain the $w$ value from observation data, we simply use $E=P-Q$ but include an uncertainty of the impact of the ignored change in water storage. To minimize the effect of change in storage, the moving average method, with a sufficient width of the moving window, is applied (Jiang et al., 2015). A moving window of 11 years was adopted by Jiang et al. (2015). In this study, the effective

width of the moving window for the basins in the SRYR of 7, 9, 11, 15 and 19 years were tentatively considered in our analysis.

## 4. Results and comparisons

### 4.1 Results of the existing separating methods

As described in Sect. 2.2, the study period between 1961 and 2013 is divided into three typical sub-

periods. The period of 1961-1990 is a reference to account for the change. For period 1991-2002, $P$ decreased (3% - 12.4%) and $E_0$ slightly increased (0%-3.3%) with time in the whole area. In the 21st Century (2003-2013), $P$ increased (1%-15.5%) across the whole study area with a sharp increase (more than 5.7%) in the JM and TNH sub-basins. $\Delta E_0$ varied from -2.1% to 7.4% in the 21st century.

Using the period of 1961-1990 as the reference period, the change in discharge was partitioned with three existing methods (climate elasticity method, sensitivity method, and decomposition method) for the three sub-basins in the two periods, 1991-2002 and 2003-2013 (Table 3). The catchment specific parameter for the climate elasticity method, $w=2$, is applied according to Zheng et al. (2009). For the sensitivity method and decomposition method, $w$ is inversely calculated from Fu's equation according

to the mean water balance in different periods. All the estimated changes in discharge are expressed in Table 3 as the percentage of the mean annual discharge during 1961-1990. As indicated, similar results are obtained from the three methods. For the 1991-2002 period, the discharge values at JM, MQ and TNH decreased more than 20%. For period in the 21st century, the discharge changed with different trends in different catchments: a slight increase trend existed at JM, and a decrease trend existed at MQ

and TNH. The anthropic changes in the discharge estimated with the existing methods, $\Delta Q_h = \Delta Q - \Delta Q_c$, was regarded as the partial of change induced by the change in the catchment specific parameter, $\Delta Q_w$ (see Sect. 3.2).

Our results show that both $\Delta Q_c$ and $\Delta Q_w$ are significant (Table 3). In general, $\Delta Q_w$ is positively related

with $\Delta Q$ and contributes more than half of the total absolute change. These results show that the change in the catchment parameter have an important role in the hydrological behaviors of the SRYR, which is similar to previous researches for the other catchments (Zheng et al., 2009, Zhao et al., 2009). A relative different result existed in the 21st century for the JM basin, where $\Delta Q$ was positive but $\Delta Q_w$ was negative and the climatic change, $\Delta Q_c$ became a dominant component in $\Delta Q$.



If $\Delta Q_w$ is completely regarded as the human-induced change in the discharge ($\Delta Q_h$), we obtain an unreasonable conclusion based on the results in Table 3; the human activities were able to cause significant negative impact on the hydrological processes in an area with a very low population density. This unreasonable conclusion indicates that something is missing from in the analysis. In fact, the catchment specific parameter depends on several catchment properties (plants, permafrost, etc.) that can be influenced by climate change such as the changes in the precipitation, evapotranspiration and temperature (Yang et al., 2007; Williams et al., 2012; Jiang et al., 2015). Accordingly, $\Delta w$ should not be completely referred to as the change in land use. Therefore, a further analysis is necessary to extract the impact of the climate change on $\Delta Q_w$.

**4.2 Results of using Eq. (16)**

In this study, we first used the new method (Sect. 3.3) to investigate $\Delta Q_w$ with the regression formula Eq. (16). The $w$ values for the three sub-basins obtained annually with different moving windows are shown in Figure 4. As indicated, an increase in the window's width can reduce the local oscillations (caused by the change in storage) of the time-$w$ curves but will also remove more variation memories by averaging all the $Q$, $P$ and $E_0$ data. As indicated by Jiang et al. (2015), when the window width is larger than 10 years, the impact of the change in storage is sufficiently small to be assumed as negligible. Thus, the 11-year window is used to retain the maximum amount of information about temporal variability without having influence from a change in storage. Using an 11-year moving average method to remove changes in water storage is further verified by GRACE data for water storage in the SRYR (details in Supplementary Material). Further, an 11-year average data were also used in previous studies (Jiang et al., 2015) and it matches the significant return period of streamflow data.

Changes in the moving average $w$ values between 1965 and 2010 are shown in Figure 5(a) together with the data points of the mean annual $w$ values in different 11-year periods. The time-varying values of $w$ in the three sub-basins significantly increased during 1980-2000. In the 21$^{st}$ century, this increasing trend continued at MQ but stopped and transitioned to a decline trend for JM and TNH. MQ has the lowest dryness index ($E_0/P$) but its $w$ value ranges between the $w$ values of JM and TNH. Using these time-varying $w$ data, the shift paths of the basin scale hydrology in the Budyko space can be plotted in Figure 5(b). Points in the Budyko space can be located respective to a given Budyko-type curve by changing the $w$ value for Fu's equation. Example classical Budyko curves for w = 2.4,2.0 and 1.7 are shown in Fig. 5(b). The gray area of Fig. 5b indicates where the Budyko type-curve changes with different $w$ values.The data points generally show a positive relationship between the evapotranspiration ratio ($E/P$) and $E_0/P$ with slopes that are larger than the slope of traditional Budyko curves with a constant parameter. This is caused by the increase in $w$ values with increasing $E_0/P$ in the study area.





The parameter $w$ in Fu's equation (Fu, 1981) represents an integrated effect of catchment properties (vegetation, soil properties and catchment topography, etc.) under a steady-state climate condition (Zhang et al., 2004; Yang et al., 2007; Williams et al., 2012). In this study, to investigate how the parameter of a basin responds to climate change, the relationship between the moving average $w$ values

and the climatic forces is analyzed with Eq. (15). Three optimal regression models (with the significance level of 0.001) were chosen through covariate analysis (Table 4). For different sub-basins, the selected variables in the respective model are not the same, but the explanatory variables contained in each model can be generally divided into two groups: water supply-related and energy supply-related. Water supply-related explanatory variables contain $I$ or $P$, and the energy supply-related

explanatory variables contain $T$ or $E_0$. According to the Budyko hypothesis, the mean annual water balance in a basin is a synthesis of water supply from precipitation and energy supply-controlled loss of water by evapotranspiration. Both water supply-related and energy supply-related variables are included in each formula (Table 4), indicating the reasonability of the obtained results.

In general, both the annual precipitation ($P$) and the annual mean precipitation intensity ($I$) are negatively correlated to the $w$ values but the correlation will be more significant when $I$ is used in the formulas. $E_0$ and $T$ are positive factors for the change in the $w$ values. For the sub-basin JM, the air temperature ($T$) seems to be a more essential role than the potential evapotranspiration ($E_0$) in controlling the $w$ value, which contrasts with that in MQ. In TNH, both $T$ and $E_0$ are significant factors

that correlated to the $w$ value. For both of JM and TNH, Figure 4 and Figure 5 clearly shows that the $w$ values of JM and TNH had an increase trend from 1990 to 2000 and a decline trend after 2000. This shift of the trends may be linked with the fact that during 1990-2000 the $T$ value significantly increased ($T$ is positively correlated with $w$) whereas after 2000 the $T$ value was relatively steady but the $I$ value significantly increased ($I$ is negatively correlated with $w$).

To assess the impact of the change on the catchment parameter on at the basin scale discharge, Equation 21 was used to calculated $\Delta Q_c$, where $\Delta Q_{c1}$ and $\Delta Q_{c2}$ were estimated with Eqs. (19) and (20), respectively. The change is related to the moving average data from 1966 with a window of 1961-1971. Results are shown in Figures 6 (a)-(c) where $\Delta Q$, $\Delta Q_c$, $\Delta Q_{c1}$ and $\Delta Q_{c2}$ are presented in the percentage of

the mean annual discharge during 1961-197. As indicated, $\Delta Q_c$ is close to $\Delta Q$ for the three sub-basins, demonstrating that the climate change is the primary cause of the discharge change in the SRYR. The absolute values of $\Delta Q_{c2}$ are generally higher than that of $\Delta Q_{c1}$, indicating that the change in the catchment specific parameter plays a more important role than the direct impacts of the changes in the precipitation and potential evapotranspiration. The contribution of the catchment specific parameter

change obtained by existing methods ($\Delta Q_w$) shown in Table 3 are similar with modified method ($\Delta Q - \Delta Q_{c1} \approx \Delta Q_{c2}$), which indicates that the modified method can obtain a time-vary results without expense of accuracy. However, existing methods attribute all catchment properties change impacts ($\Delta Q_w$) to human activities ($\Delta Q_h$). In JM and TNH, as shown in Figures 6 (a) and (b), the absolute $\Delta Q_{c2}$





value was higher than twice of the absolute $\Delta Q_{c1}$ value between 1988 and 2006. However, the anthropic change in discharge could not be sufficiently estimated with $\Delta Q_h = \Delta Q - \Delta Q_c$ from these results, because the impact of the permafrost was not considered in Eq. (16).

### 4.3 Results of using Eq. (17)

The regression formula, Eq. (17), is more appropriate in comparison with Eq. (16) in this study because the study area is a cold region with permafrost. Substituting the results of Eq. (16) into Eq. (15), we obtain the residual part of $w$ as $X$. The need for using Eq. (17) can be indicated by the correlation between $X$ and MFD, as shown in Figure 7. It is clear that $X$ is positively correlated with MFD in JM ($R^2 = 0.58$, significance level $<0.001$) and in TNH ($R^2 = 0.47$, significance level $<0.001$). However, this

$X$-MFD correlation is very weak in MQ. A higher average value of the MFD seems lead to a stronger correlation between the $X$ and MFD.

According to the significant correlation between $X$ and MFD in JM and TNH, a further stepwise regression analysis is carried out using Eq. (17) for JM and TNH, which leads to

$$w = 4.002 + 0.557\overline{T} - 2.237\overline{I} + 1.072\overline{D} \text{, for JM} \tag{26}$$

$$w = -1.782 + 0.523\overline{T} - 2.498\overline{I} + 4.665\overline{E}_0 + 0.954\overline{D} \text{, for TNH} \tag{27}$$

These regression formulas can increase the correlation coefficient to $R^2 = 0.92$ and $R^2 = 0.93$ for JM and TNH, respectively. The values of coefficient $\beta_4$ are positive and close to 1.0, indicating that MFD is a positive factor for the catchment parameter. Since MFD decreases with the degradation of permafrost,

this positive relationship between $w$ and MFD implies that the degradation of permafrost will cause negative changes in the $w$ and $E/P$ values and then increase the discharge in JM and TNH. Eqs. (26) and (27) also show that the $w$ value is positively correlated with the air temperature, $T$, and negatively correlated with the precipitation intensity, $I$, in both sub-basins and is positively correlated with $E_0$ in TNH. These results of the relationship between the catchment parameter and the normal climatic forces

are consistent with that shown in Table 4, but the values of the coefficients change.

Based on Eqs. (26) and (27), the results of $\Delta Q_{c1}$, $\Delta Q_{c2}$ and $\Delta Q_c$ were estimated again with Eq. (17) for JM shown in Figure 6 (d) and TNH shown in Figure 6 (e). The moving average data of MFD for the sub-basins are limited to the period between 1966 and 2000 so that curves after 2000 are not shown in

Figure 6 (d) and 6 (e). In comparison with the results of Eq. (16) shown in Figures 6 (a) and 6 (b), the difference between $\Delta Q$ and $\Delta Q_c$ in Figures 6 (d) and 6 (e) obviously decreases because most of the change in discharge are attributed to the climatic change. However, the time-varying patterns of $\Delta Q$ and $\Delta Q_c$ do not change much.




The anthropic derived change in discharge attributed is now estimated with $\Delta Q_h = \Delta Q - \Delta Q_c$ from the new results of $\Delta Q_c$, as shown in Figure 8. As indicated, the anthropic change in discharge is generally less than 5% of the total change. Before 1990, the $\Delta Q_h$ values appear to show a random oscillation, which may not really indicate the anthropic change. After 1990, the $\Delta Q_h$ values in JM and TNH shown a long-

term increase trend whereas the $\Delta Q_h$ values in MQ shown a negative trend. These trends may be linked with the environment conservation program implemented in the 1990s the SRYR but it is not clear for that why MQ shown different trend from that in JM and TNH.

## 5 Discussions

### 5.1 Controls of the change in MFD

In Sect. 4.3 we found that the catchment parameter ($w$) in JM and TNH is correlated with MFD and roles as an influence factor for the discharge in additional to the normal climatic factors. Questions arise here with the relationship: what controls the change in MDF? Is the MDF value depends on the air temperature that have been included in Sect. 4.2?

It was well known that global warming has triggered degradation of the cryosphere, including glaciers and permafrost in the Tibetan Plateau. Such a relationship is exhibited in Figure 2 where MFD generally decreased in the study period whereas the air temperature shown an increase trend. To analyse this correlation, we compared the MFD data for 11 stations with the annual mean air temperature (MAT) and the cold season (from October to March) mean air temperature (CMAT).

Locations of these stations are shown in Figure 1b and the correlation analysis results are shown in Table 5. In general, the significance of the correlation between MFD and CMAT is higher than the correlation between MFD and MAT, except for two stations (M8, M11) where the record lengths were less than 15 years. This indicates that the decrease in MFD may be controlled by the increase in the air temperature in the cold seasons. However, the R-squared correlation coefficients are not high,

indicating that the region may be influenced by other factors such as the soil temperature and moisture, snowpack and canopy (Goodrich, 1982; Nelson et al., 1997; Yamazaki et al., 1998; Shanley and Chalmers, 1999). However, at present it is difficult to quantitatively assess the impacts of these factors because the lack of data.

If the negative correlation between the MFD and MAT is significant, it will influence the meaning of the regression formulas for the catchment parameter, $w$, as is presented in Table 4 and Eqs. (26) and (27). According to Eqs. (26) and (27), $w$ can be determined by

$$w = w_f + \beta_3 \overline{T} + \beta_4 \overline{D} \qquad (28)$$





where $w_f$ is the components related to the other factors, $\beta_3$ and $\beta_4$ are the coefficients for the MAT and MFD (both are positive), respectively. Further, we can describe the MFD-MAT correlation as

$$\overline{D} = -\xi\overline{T} + \eta \qquad (29)$$

where $\xi$ is the absolute slope of the correlation line and $\eta$ is the residual resulting from other controls.

Substituting Eq. (29) into Eq. (28) yields

$$w = \left(w_f + \beta_4\eta\right) + \left(\beta_3 - \beta_4\xi\right)\overline{T} \qquad (30)$$

Because $\beta_3$, $\beta_4$ and $\xi$ are positive coefficients, then $\beta_3 - \beta_4\xi$ must be less than $\beta_3$. Accordingly, the MFD-MAT correlation may reduce positive sensitivity (when $\xi$ is less than $\beta_3/\beta_4$) or cause negative sensitivity (when $\xi$ is higher than $\beta_3/\beta_4$) of the catchment parameter to the change in the air temperature.

This may explain why the coefficient of $T$ in Table 4 is less than that of Eqs. (26) and (27).

If $\beta_3 \approx \beta_4\xi$, then the role of MAT cannot be explicitly included in Eq. (30). This may be the reason why $T$ is not included in the formulas of the sub-basin MQ in Table 4 though the effects of the air temperature should also exist in the basin. Such an effect could also be applied to explain why the relation between $X$ and MFD is weak in MQ (Figure 7). It is speculated that in MQ the $w$ value is also positively correlated with MFD but is offset by the effect of air temperature. In general, the increasing air temperature will increase evapotranspiration and decrease the discharge in catchments without permafrost (Yang and Yang, 2011; Tang et al., 2012). However, in the cold Tibetan plateau region, the degradation of permafrost due to increasing air temperature can enlarge the discharge of groundwater

and partially counteract the negative impact of the increasing air temperature (McKenzie *et al.*, 2013). This particular effect leads to a more complex response of streamflow in the cold basins with permafrost than that in warm basins without permafrost.

### 5.2 Limitation remarks

This study does not directly consider permafrost degradation as a major research objective in

understanding the linkages between climate change and runoff generation at the catchment scale. It is different from previous studies which regard permafrost as a main research object with subsurface domain. Most of the previous research about permafrost degradation highlight temperature increase impacts on permafrost degradation, leading to changes in soil physical properties, groundwater storage, and discharge. However, these changes are focused on the subsurface domain, rather than at the

catchment scale. In these previous studies, changes in the interactions between subsurface and surface conditions (e.g. evaporation, precipitation, and precipitation intensity) are neglected. Considering the complex impact of permafrost degradation at the catchment scale on the hydrological system, it is necessary to account for this process to better understand and predict change. At present, there are few physical models that can accurately assess these situations; we demonstrate that these changes can be

preliminarily considered using conceptual models and statistic methods. An important finding from our





research is that by considering permafrost at the catchment hydrological scale, degrading permafrost is a positive factor for discharge change.

In this study, we highlight the role of the permafrost degradation with global warming in the change of discharge in the SRYR because the catchment parameter ($w$) is positively correlated with the MFD. An
increasing trend in MFD implies a decreasing trend in permafrost. However, the MFD is not a full measurement on the thickness (or volume) of the permafrost which only represents the surficial part of the permafrost. We could not avoid this limitation because the direct observation data of the varying thickness for the permafrost are not available. Observation of the MFD is also difficult and limited in a few observation stations. In each of the three studied basins, we chose one observation station to
extract the representative MFD data for the correlation analysis. This is sufficient to capture the general relationship but may induce quantitative errors because the distributed patterns of the MFD are not considered. It is possible to use several geophysical methods in characterizing distributed frozen depth of shallow soils, such as: electrical resistivity imaging, electromagnetic induction, ground penetrating radar and infrared imaging (Briggs et al., 2016). However, these methods could only be performed in a
small area (the size is generally less than 1 km) and are too expensive to be applied everywhere within a larger catchment.

An unresolved question is why does the decreasing MFD act as a positive factor for the discharge of basins in the SRYR? This is still an unsolved problem that was not resolved with only the Budyko
framework.  It has been found that the permafrost degradation could enlarge baseflow in cold regions (Walvoord and Striegl, 2007; Bense et al., 2012; Evans et al., 2015;). Decrease in MFD because of global warming was considered as a major factor for the increase in baseflow in the Qilian mountain, China (Qin et al. 2016). Further deeper investigations are required to link the rainfall-runoff and base flow behaviors with the physical mechanism of frozen soils and performed in the SRYR.

There are also potential limitations in the data processing. We extracted the annual data of $P$, $E_0$, $T$ and $I$ for the three sub-basins from the gridded data of IDW interpolation. Among them, the precipitation data agree with the TRMM 3B42 data, indicating the efficiency of the IDW method. However, it is difficult to assess how the elevation-dependent variations of $P$, $E_0$, $T$ and $I$ were sufficiently included in
the IDW interpolation. Quantitative analyses of the elevation effects on these meteorological factors are very limited in the literature. The elevation effects could be partly dealt with many distributed weather stations (Figure 1b). The influences of the local topographic relief of the ground surface in a sub-basin were expected to be counteracted at the catchment scale and nonsignificant for the time-varying patterns. Nevertheless, it is an essential problem which requires much more investigation in
further studies. Another problem involves the normalization of data in the regression analysis. In Eqs. (15)-(17), $\overline{T}$ ($=T/T_m$) is normalized with $T_m$ where $T_m$ are not close to zero. This is because a very small $T_m$ value could significantly enlarge the relative change in $T$ and increase the errors in estimating $\Delta w$.



Since the objective is to estimate a dimensionless change, a larger value of $T_m$ can be used if the mean annual air temperature is close to zero.

## 6. Conclusions

This work modifies and improves on the traditional separation approach to identify the contribution of
climate change to the change in discharge of drainage basins. This modified approach is applied for the source region of the Yellow River (SRYR) in analyzing the inter-annual change in discharge for a past 50 years. The role of varying permafrost in this cold region is investigated by using the observation data of the maximum frozen depth. According to the obtained results, several conclusions can be highlighted:

(1) The change in discharge of the SRYR in the past half-century is mainly controlled by the climate change rather than the local human activities. In the 1990s, a significant decrease in discharge was triggered by the increase of the air temperature and the decrease of the annual precipitation. In comparison with the annual precipitation, the precipitation intensity is a stronger control of the discharge in the SRYR.

(2) Analysis of the inter-annual water balance with Fu's formula in the Budyko framework shows a positive relationship between the catchment specific parameter and the aridity index. The change in the catchment specific parameter, $w$, is interpreted to be due to climate change and not land use change, differing from previous research. Moreover, in two of the three studied basins the catchment specific parameter is correlated with the maximum frozen depth of permafrost, which indicates the impact of
the permafrost on discharge behaviors.

(3) The negative correlation between the maximum frozen depth and air temperature indicates the impact of global warming on permafrost degradation but implies conflicting effects of the air temperature rising on the discharge. The increasing air temperature is generally a negative force for discharge. However, in our cold study area it also causes degrading permafrost that can act as a positive
factor for discharge. Such conflicting effects enhance the uncertainty in assessments of the hydrological response to climate change in the SRYR.

### Acknowledgements

This research was supported by the China Geological Survey from 2000, the National Natural Science Foundation of China (41330634, 41072191, 91125011) and the Fundamental Research Funds for Central Universities. Acknowledgement for the data support from "Loess Plateau Data Center, National Earth System Science Data Sharing Infrastructure, National Science & Technology Infrastructure of China (http://loess.geodata.cn).



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




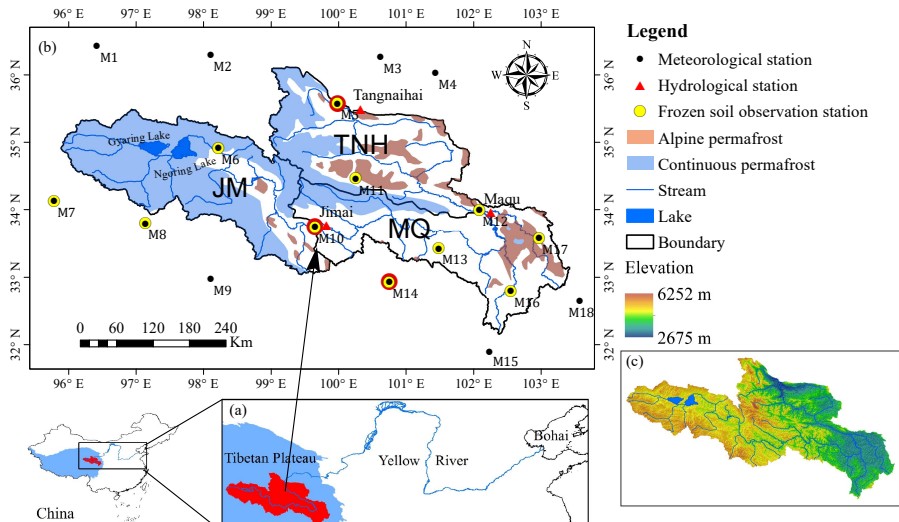

**Figure 1 (a) Location map of the source region of the Yellow River (JM, MQ and TNH) (b) Distribution of permafrost, meteorological and hydrological stations (The red circles indicates stations which have the longest frozen soil record in the respective sub-basin.). (c) Elevation of the study area.**





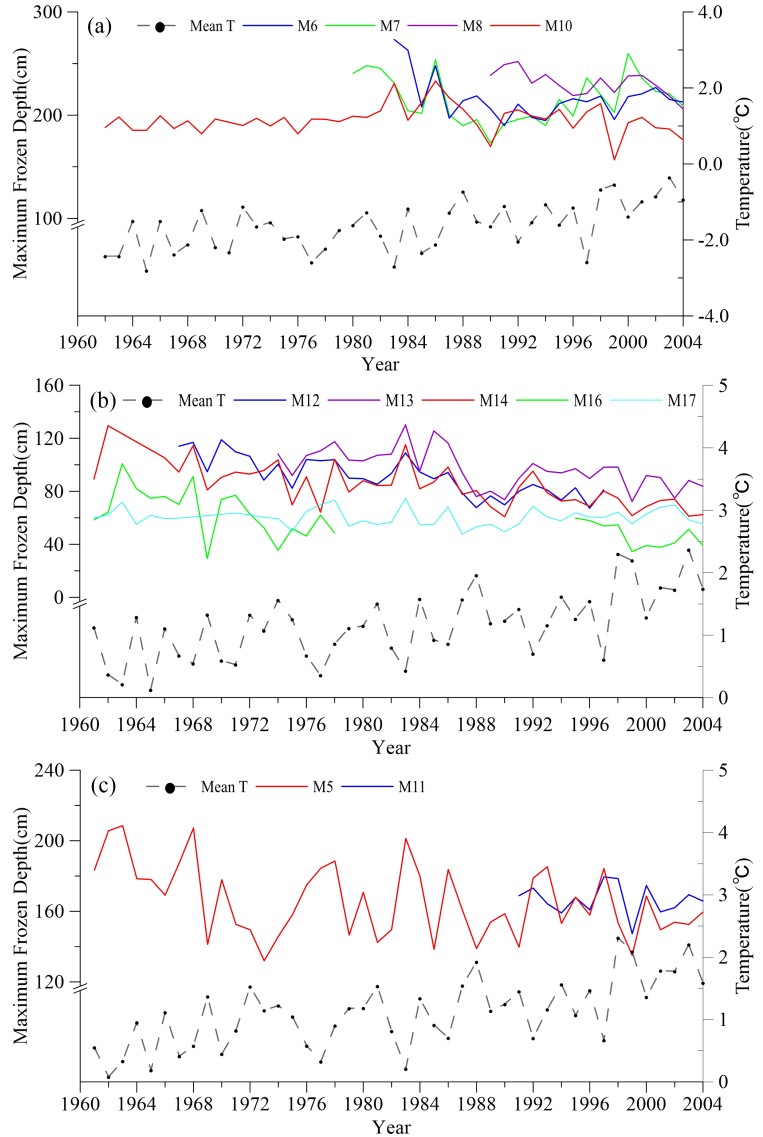

**Figure 2 Change in the maximum frozen depth (MFD) and annual mean temperature of (a) JM, (b) MQ**
5  **and (c) TNH subbasins.**




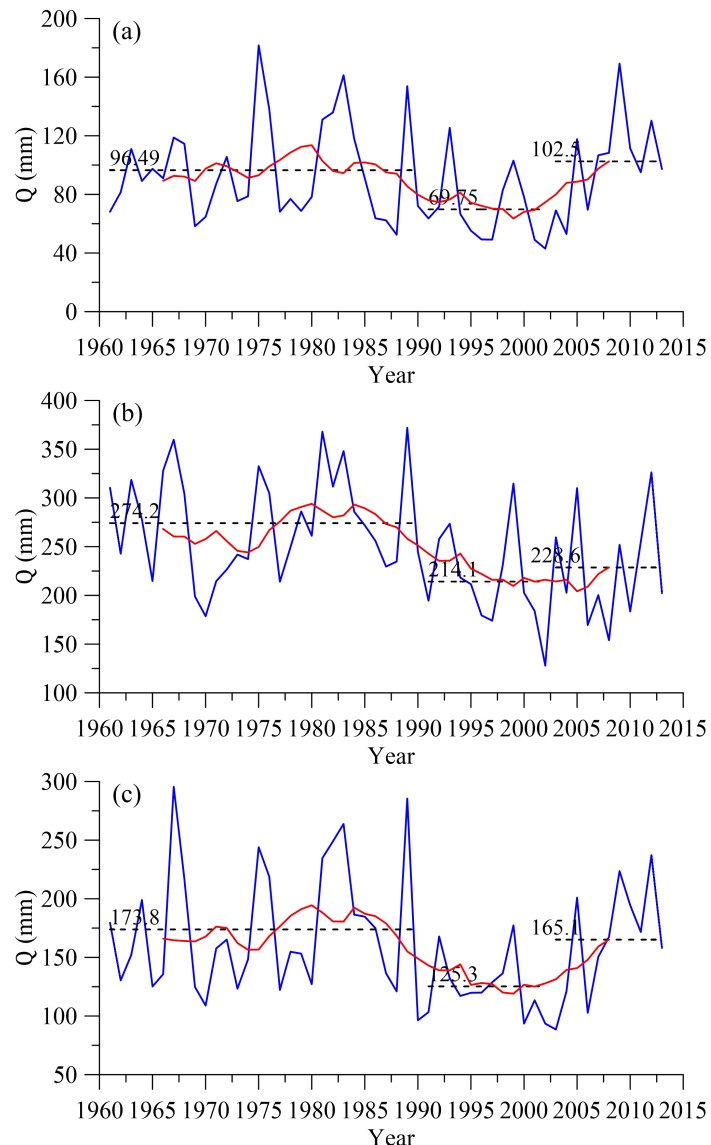

**Figure 3 Time series of annual discharge, _Q_, in (a) JM, (b) MQ, and (c) TNH. The dashed line, red line and blue line indicate the mean value in different periods, 11-year moving average value and annual mean value of discharge, respectively.**





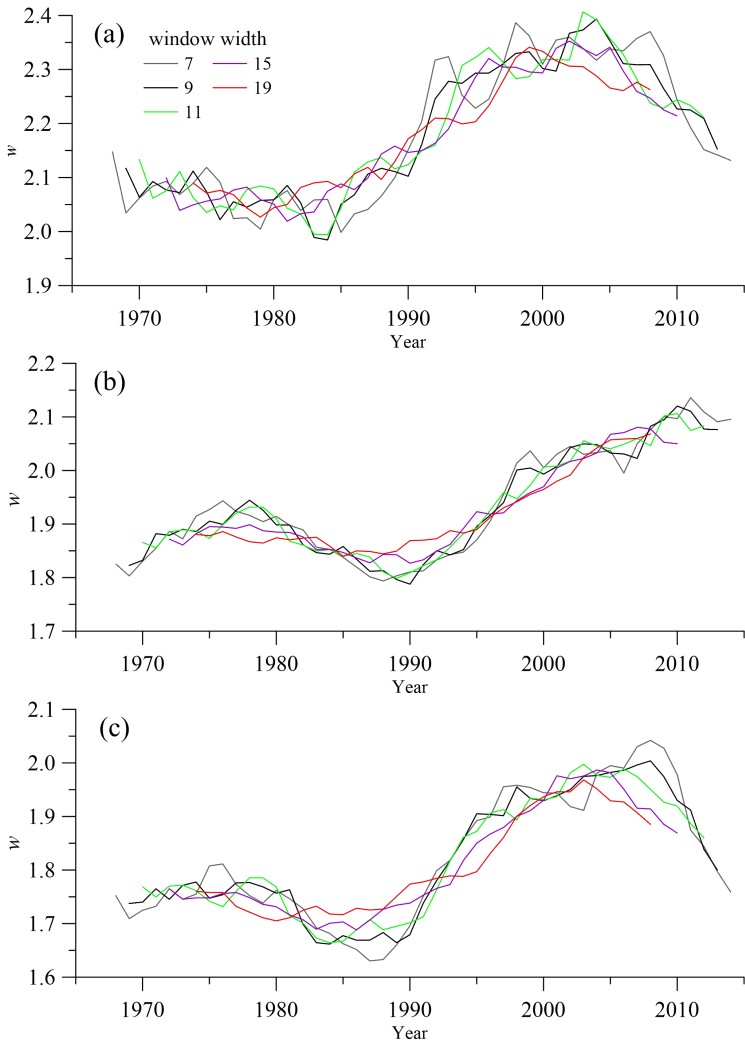

**Figure 4 Time series of the catchment parameter (w) obtained from different widths of the moving windows for JM (a), MQ (b) and TNH(c).**




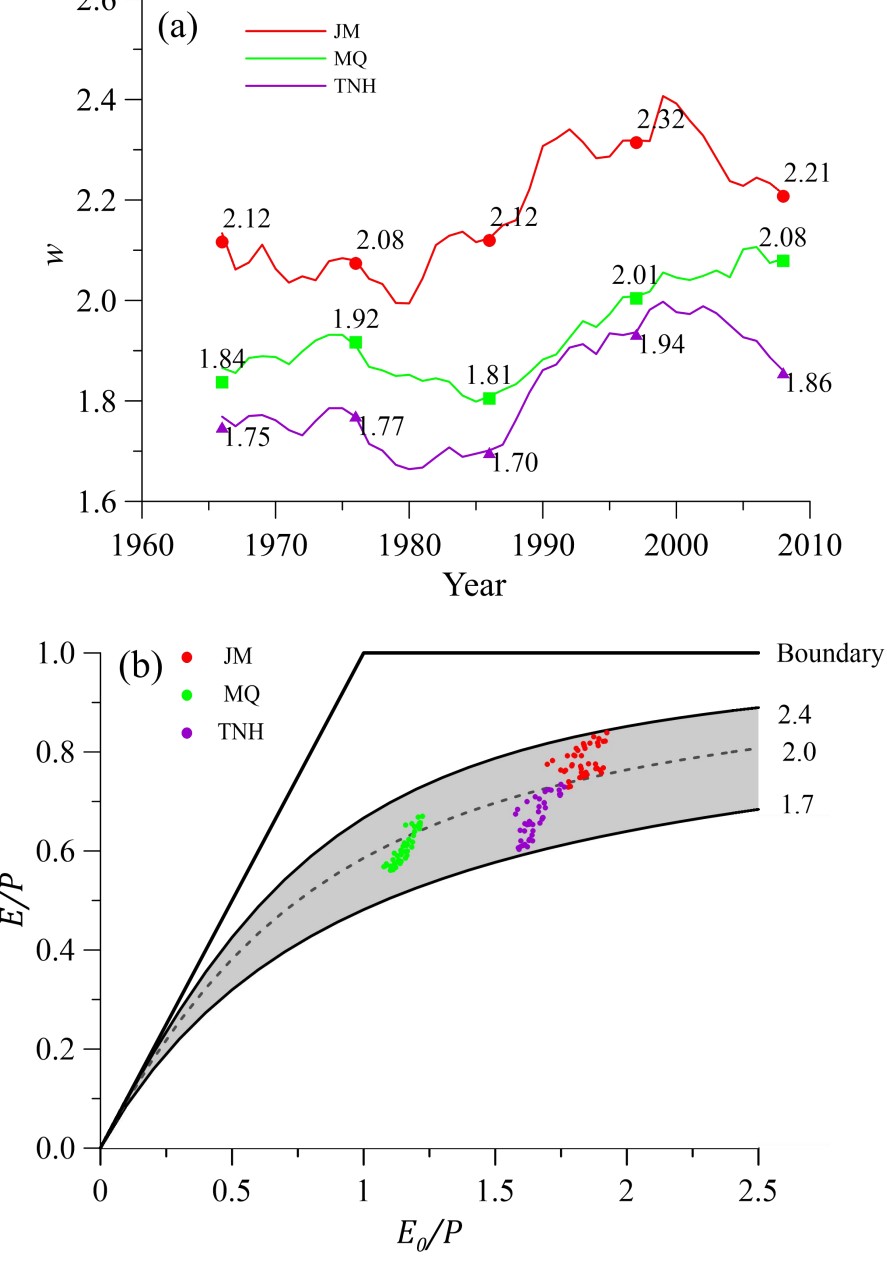

**Figure 5 Time-varying values of the catchment parameter ($w$) in Fu's formula (a) and plots of the moving average water balance in the Budyko space (b) for the three sub-basins. The labelled points in (a) show the mean annual $w$ values for different periods (1961-1970, 1971-1980, 1981-1991, 1992-2002, 2003-2013) which are plotted for the mid-point of the time periods. In (b), 2.4, 2.0 and 1.7 refer to the $w$ value for the three type curves.**





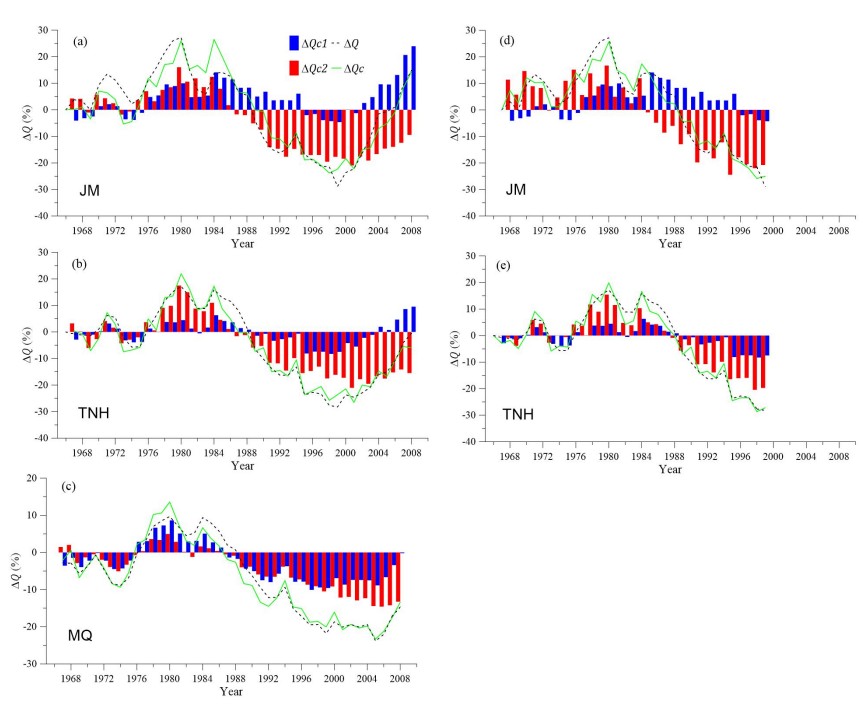

Figure 6 Decomposition results of the climatic change in discharge ($\Delta Q_{c1}$, $\Delta Q_{c2}$ and $\Delta Q_c$) in comparison with the total change in discharge ($\Delta Q$) for the sub-basins. (a), (b) and (c) are the results of using Eq. (16) for JM, TNH and

10   MQ, respectively. (d) and (e) are the results of using Eq, (17) for JM and TNH, respectively.



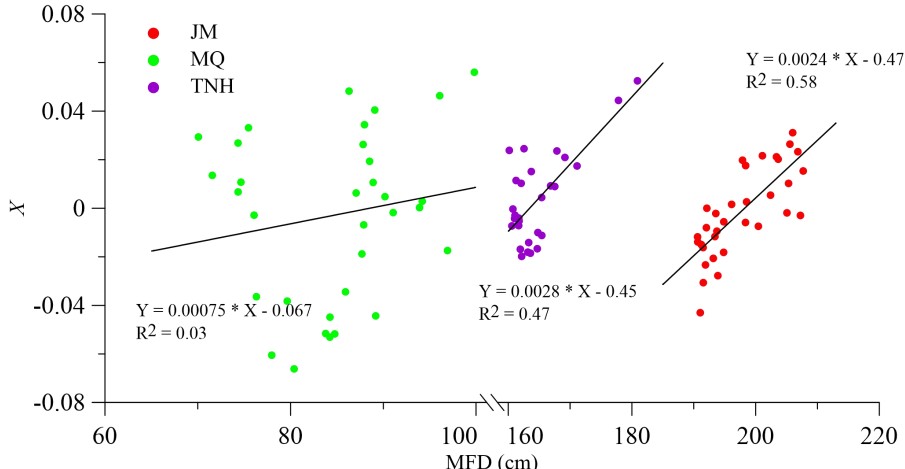

**Figure 7  Correlation between X and the maximum frozen depth (MFD). The X values are estimated using Eqs. (15) and (16).**

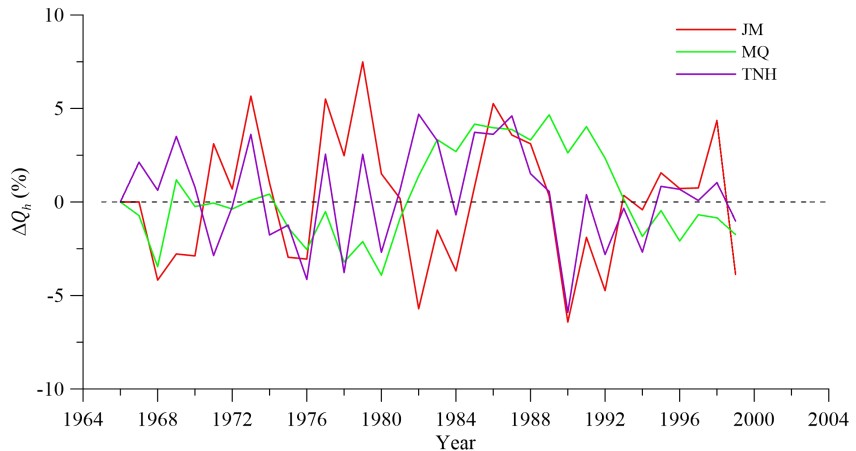

**Figure 8 Time series of $\Delta Q_h$ that obtained by using Eq. (17)**



Table 1 Previous study of discharge change of the Yellow River

| Main types of study | | Data/Method used | Study area | Main causes of discharge change |
|---|---|---|---|---|
| Data analysis | Yang et al.(2004) | Data from 108 meteorological stations and irrigation data | The Yellow River basin | Decrease in precipitation and increase in evapotranspiration |
| | Zheng et al. (2007) | Non-parametric Mann–Kendall test and flow duration curve | Sub-basins above Tangnaihai station | Decreasing of wet-season rainfall |
| | Liang et al. (2010) | Wavelet method | Sub-basins above Huangheyan Station | Climatic variations. |
| Process based modeling | Tang et al. (2007) | DBD model | The Yellow River basin | In the upper and middle reaches: climate change |
| | Tang et al. (2013) | SWAT | Sub-basins above Huayuankou station | Precipitation decrease in 1990s |
| | Cuo et al. (2013) | VIC | Sub-basins above Jingyuan County | Climate change is primary cause for discharge change above Tangnaihai station |
| | Meng et al. (2016) | VIC | Sub-basins above Tangnaihai station | Changes of precipitation and evapotranspiration |
| Climate elasticity | Zheng et al. (2009) | Climate elasticity based on Budyko type equations | Sub-basins above Tangnaihai station | Land use change |
| | Zhao et al. (2009) | Climate elasticity based on Zhang's equation | Sub-basins above Lanzhou station | Climate change and human activity |





Table 2 Proposed formulas of $E/P$ versus $E_0/P$ in the Budyko framework

| Types | Solutions | Equations |
|-------|-----------|-----------|
| Non-parametric formulas | Schreiber (Schreiber,1901) | $1 - e^{\frac{E_0}{P}}$ |
| | Ol'dekop (Ol'dekop,1911) | $\frac{E_0}{P}\tanh\left(\frac{P}{E_0}\right)$ |
| | Budyko (Budyko, 1974) | $\left[\frac{E_0}{P}\tanh\left(\frac{P}{E_0}\right)\left(1 - e^{-\frac{E_0}{P}}\right)\right]^{\frac{1}{2}}$ |
| One-parameter formulas | Fu (Fu,1981;Zhang et al., 2004) | $1 + \frac{E_0}{P} - \left(1 + \left(\frac{E_0}{P}\right)^w\right)^{\frac{1}{w}}$ |
| | Choudhury (Choudhury,1999; Yang et al., 2008) | $\left(1 + \left(\frac{E_0}{P}\right)^{-n}\right)^{-\frac{1}{n}}$ |
| | Zhang (Zhang et al., 2001) | $\dfrac{1 + \alpha \cdot \frac{E_0}{P}}{1 + \alpha \cdot \frac{E_0}{P} + \frac{P}{E_0}}$ |





Table 3 Results of partitioned changes in discharge using existing methods

| Sub-basins | Periods | $w$ | | | $\Delta Q_c$ (%) | | | $\Delta Q_h (\Delta Q_w)$ (%) | | | $\Delta Q$ (%) |
|---|---|---|---|---|---|---|---|---|---|---|---|
| | | C* | S* | D* | C | S | D | C | S | D | |
| JM | 1961-1990 | 2 | 2.10 | 2.10 | | | Reference period | | | | |
| | 1991-2002 | 2 | 2.33 | 2.33 | -8.10 | -7.25 | -7.69 | -19.62 | -20.67 | -20.03 | -27.72 |
| | 2003-2013 | 2 | 2.21 | 2.21 | 19.57 | 16.37 | 18.60 | -13.35 | -10.05 | -12.38 | 6.22 |
| MQ | 1961-1990 | 2 | 1.85 | 1.85 | | | Reference period | | | | |
| | 1991-2002 | 2 | 2.01 | 2.01 | -11.04 | -11.20 | -11.43 | -10.85 | -10.74 | -10.47 | -21.89 |
| | 2003-2013 | 2 | 2.08 | 2.08 | -1.29 | -1.35 | -1.18 | -15.31 | -13.52 | -15.42 | -16.60 |
| TNH | 1961-1990 | 2 | 1.74 | 1.74 | | | Reference period | | | | |
| | 1991-2002 | 2 | 1.94 | 1.94 | -8.20 | -8.89 | -9.37 | -19.75 | -19.25 | -18.58 | -27.94 |
| | 2003-2013 | 2 | 1.86 | 1.86 | 6.84 | 7.34 | 8.38 | -11.86 | -11.05 | -13.40 | -5.02 |

* The C, S and D indicate the climate elasticity method, sensitivity method and decomposition method, respectively.

Table 4 Results of $w$ covariate analysis for the period between 1961 and 2013

| Sub-basins | Models of Eq. (15) (Candidates $\overline{P}$, $\overline{E_0}$ and $\overline{T}$) | $R^2$ | Models of Eq. (15) (Candidates $\overline{I}$, $\overline{E_0}$ and $\overline{T}$) | $R^2$ |
|---|---|---|---|---|
| JM | $4.254+0.357\overline{T}-1.715\overline{P}$ | 0.67 | $5.229+0.466\overline{T}-2.582\overline{I}$ | 0.83 |
| MQ | $-1.743-1.201\overline{P}+4.874\overline{E_0}$ | 0.85 | $-3.778-1.582\overline{I}+7.291\overline{E_0}$ | 0.88 |
| TNH | $3.814+0.287\overline{T}-2.281\overline{P}$ | 0.90 | $1.982+0.442\overline{T}-3.194\overline{I}+2.591\overline{E_0}$ | 0.91 |





Table 5 MFD record length and correlation between CMAT and MAT at observation stations

| Station | $R^2$ of MFD- CMAT | $R^2$ of MFD- MAT | MFD record length (years) |
|---------|-------------------|-------------------|---------------------------|
| M5 | 0.65 (–)* | 0.49 (–) | 1960-2004(45) |
| M6 | 0.29 (–) | 0.13 (–) | 1983-2004(22) |
| M7 | 0.34 (–) | 0.053 (–) | 1980-2004(25) |
| M8 | 0.18 (–) | 0.41 (–) | 1990-2004(13) |
| M10 | 0.21 (–) | 0.08 (–) | 1962-2004(43) |
| M11 | 0.01 (–) | 0.18 (–) | 1991-2004(14) |
| M12 | 0.55 (–) | 0.46 (–) | 1967-1997(31) |
| M13 | 0.58 (–) | 0.41 (–) | 1974-2004(31) |
| M14 | 0.47 (–) | 0.24 (–) | 1966-2004(39) |
| M16 | 0.47 (–) | 0.38 (–) | 1971-1979  1995-2004(28) |
| M17 | 0.52 (–) | 0.35 (–) | 1974-2004(31) |

5        *(–) indicates negative correlation