# Peer review of "Climate-induced hydrologic change in the source region of the Yellow River: a new assessment including varying permafrost"

_Hydrology and Earth System Sciences, 2017_

## Referee Comment (RC1) · Anonymous Referee #1 · 24 Feb 2018

**General comments**

Wu et al. conduct an analysis of long term climate and discharge data in the source region of the Yellow River to attempt to disentangle the different factors contributing to discharge changes. Their motivation for doing so, in additional to the obvious societal important of the Yellow River, is that past studies seemingly attribute too high a portion of the discharge sensitivity (or at least discharge changes) to anthropogenic activity despite the relatively low population density. In essence, they modify the classic Budyko approach to demonstrate that changes to the hydrologic framework due to permafrost thaw (as measured by changes in the maximum frozen depth, MFD) produced

by climate change are more likely to be dominant drivers in the discharge changes, than anthropogenic activity per se (although one might argue that the global-scale anthropogenic activity actually produced the climate change). These complicating effects result in offsetting discharge forcings and thereby increase uncertainty. Generally, the paper is understandable, reasonable, and it moves the field forward. I think parts of it could be communicated better and these modifications should be made before it is seriously considered for publication.

**Major comments**

1. In dozens of places there are grammatical errors, missing punctuation, or improper (or at least less than idea) word usage. I'd encourage the authors to have an English colleague read the paper carefully or to go through an English editing service. It is not in terrible shape, but it could be improved.

2. Duan et al. (2017) tackle a similar problem to the present study but in a different approach. I'd encourage the authors to explain why their approach is superior, or at least why it might be preferred in some cases. This could occur in the intro or in the discussion perhaps. In general, the authors should be highlighting that climate change can result in not just differences in the forcing, but also in the system itself (e.g. climate change can alter precipitation regimes which is the hydrologic loading, but it can also alter the hydrologic functioning of a landscape by altering the vegetation and permafrost or seasonally frozen ground distribution)

Duan et al. 2017. Distinguishing streamflow trends caused by changes in climate, forest cover, and permafrost in a large watershed in northeastern China. Hydrol. Process. 31

3. The introduction is generally ok, but I do not like the build up to the objectives at the end. For example, the authors state (P3, L10-13): "these relationships have not been previously examined in the SRYR". So is this just a case study then? If so, perhaps HESS would not be interested in publishing it. I would rather argue that the

paper advances the science by further breaking down these offsetting or cumulative discharge disturbances, and specifically examine the role of permafrost degradation – which has been largely ignored in past studies using these statistical approaches.

4. P5, L5, The daily frozen depth of the active layer was identified every month. This makes no sense. How do you identify something with daily frequency on a monthly basis? Also, here the authors say active layer, but is there truly an active layer everywhere – in many places there is no permafrost, correct? Anyway, this is all worded confusingly. The "frozen depth of the active layer" could mean the distance from the land surface to the bottom of the frozen zone (which unless there is a horizontal talik would mean all the way to the bottom of the permafrost during the winter) or it could mean the distance from the land surface to the top of the frozen zone, which would be better called the thawed depth. I'd suggest this be reworded

5. Figure 3 and P5, L20-25: How were these change points detected? Was any sensitivity analysis conducted on moving these change points incrementally forward or backward in time?

6. The transition from Eq. 7 to Equations 8-10 was hard for me to follow. This should be reworded

7. P8, L14-17, a key message of this paper is that climate change can influence the catchment specific parameter. This is likely true, but it should be explained rather than just citing a few other studies. If this is so important, the authors should provide some examples of why this might be from a physical perspective.

8. Related to the above, the paper would be much improved by the authors tying in the physical hydrologic environment to the statistical analysis results. For example, why would rainfall-runoff processes be altered by changes in permafrost, particularly in the source region of the Yellow River. What soil is there? What slope? Based on this why might the runoff ratios change? Without tying the results to the physical setting, the entire results section comes across as a bit of an arm waving exercise.

СЗ

**Minor comments**

I'd invite the authors to refer to Wang et al. 2018, which has some overlap with the present study (especially geographically) along themes related to changing MFD. They may be able to feed some data from this prior paper into their statistical approach. Wang et al. 2018. Historical and future changes of frozen ground in the upper Yellow River Basin. Global and Planetary Change, 172, 199-2011

P1, L29 'are facing serious water shortages' – use of the present tense here might warrant a more recent citation as Yang et al. (2004) is now 14 years old.

P3, L7-8, 'permafrost thawing to surface water discharge' - I don't really like this wording. It seems to imply that permafrost thaw produces streamflow (i.e. the meltwater is a significant contributor to streamflow). If so, that would surely be an incredibly fast thaw rate!

P3, L30-40, I find this description confusing. The opposite ends of the permafrost spectrum are continuous and isolated, so why would the authors lump those into a single 'continuous' category. Also Figure 1 shows alpine permafrost, but does not indicate if this is continuous or discontinuous. I guess that is what the authors are explaining in that paragraph – the classification system is not standard

P4, L3, Why is the SRYR unique? This is not explained

P17, L15, what about aerial geophysical methods in permafrost? See Minsley et al. Minsley et al. 2012, Airborne electromagnetic imaging of discontinuous permafrost. Geophys. Res. Lett.

P17, L18-24, this paragraph is worded as though it were a key springboard to future work. I found the logic in the section hard to follow. Perhaps the wrong word is used in some sentence, or perhaps my mind is dense after another long day. But they seem to suggest that decreasing MFD is a positive factor and that this is enigmatic – but then they indicate later that Qin et al. 2016 showed this. Are they saying there are no

physical explanation for this or that it is unusual?

The figures are generally well done and interesting

---

## Referee Comment (RC2) · Anonymous Referee #2 · 4 Mar 2018

**Comments to Authors:**

The manuscript entitled "Climate-induced hydrologic change in the source region of the Yellow River: a new assessment including varying permafrost" by Wu et al. (2017) used the Budyko's framework to separate the effects of climate change, human activities, and permafrost degradation on streamflow. The main findings are: 1) climate change played an important role in streamflow variations, and 2) degrading permafrost can act as a positive factor for streamflow. The endeavours in adopting Fu's equation in accounting the permafrost contributions to hydrological cycle is encouraged, and further this is an emerging research topic recently. However, a recently published paper, Wang et al. (2018) used the Yang-Choudhury equation (a Budyko based equation) to quantify climate change, land cover change, and permafrost degradation on streamflow in the same watersheds during the same study periods. One of the main conclusions of Wang et al. (2018) is that frozen ground degradation could reduce streamflow. After an assessment of two studies, I found that Wu et al. (2017) misinterpreted the Fu's and Budyko's framework, which led to contrasting results with Wang et al. (2018). I have to, therefore, reject this manuscript for further publication. Here are main comments that may help authors improve their study.

**(a) Application Budyko hypothesis**

The Budyko's hypothesis and its several frameworks, e.g., Fu and Yang-Choudhury equations are receiving considerable attention in the recent decade. They are developed based on the water balance equation, in which the $\Delta$s or water storage changes are usually neglected in the areas with no permafrost coverage (Please change your express in Page 6 line 6, as $\Delta$s can be either positive or negative). However, the degradation of permafrost can either increase (Duan et al., 2017) or decrease streamflow (Wang et al., 2018). As such, the permafrost act as either an extra source (another source of water input in the watershed) or net loss (recharging groundwater). Either scenario would likely lead to significant changes in $\Delta$s. In your study watersheds, the inter-annual changes of $\Delta$s can be reached to more than 10% of precipitation in your study watersheds (Figure 3 in Wang et al., 2018). In this case, the $\Delta$s should be seriously accounted in the water balance equation. Thus, three methods used in your study (climate elasticity method, sensitivity method, and decomposition method) are not appropriate as they are all developed based on watershed without permafrost coverage. In contrast, Wang et al. (2018) considered $\Delta$s and provided robust inferences to support their conclusion. Therefore, I think neglecting the $\Delta$s is the main reason for the contrasting results.

**(b) Watershed property parameter (w) in the Fu's equation**

The authors also tried to employed watershed property parameter (w) to identify the climate impacts. The multiple linear equations only including climate variables suggested by Jiang et al. (2015) that is insufficient. It should be noted that Jiang et al. (2015) used this equation in the non-permafrost region. Hence, it is still questionable to apply such method in the permafrost region. In addition, this application needs to be further modified. The w is closely related to the watershed slope (Yang et al., 2017; Zhou et al. 2015), vegetation (Wei et al., 2018; Zhang et al., 2018), soil properties (Yang et al., 2007; Wang and Alimohammadi, 2012), and climate (Berghuijs et al., 2017; Zhang et al., 2016 and 2018). Especially, Yang et al. (2007) and Wang and Alimohammadi, (2012) revealed that watershed balance is closed related to relative infiltration capacity, relative soil and water storage, and average slope. Moreover, you may check the vegetation change in your watersheds, Wang et al. (2018) revealed that LAI increased during your study period, which could potentially be a negative factor to explain the decrease in streamflow in your watersheds.

**(c) Comments on writing**

Overall, the manuscript is in moderate shape. However, the literature review is not sufficient. In the introduction, your put much efforts on explaining your hypothesis that human activity is a minor

effect in your study area. You can explain something more in the current research gaps in permafrost regions and highlight the uniqueness of your study. After your reading of the listed references, you may find that Table 1 has been listed in many references. In Methods sections, you introduced details about three methods. However, those three methods and Budyko frameworks have been well-documented in the literature (Dey and Mishra, 2017). This is not necessary. As you mentioned the permafrost in the manuscript title, I suggest you can discuss more in your future revision.

**(d) Suggestions for revision**

It is good to see this manuscript using Fu's equation to advance our knowledge in understanding climate, human activity, and permafrost thawing, and also advance Fu's application. As I stated previously, the author should try to close the watershed balance in your revision before applying Fu's equation. I suggest author may use analytical solutions to account for permafrost degradation contribution to either increase or decrease streamflow. Please see an example in Duan et al. (2017) and others. Then, using the modified streamflow in Fu's equation to address your research questions. As I can see from your paper, the best thing is that you have the long-term observed maximum frozen depth in your study region, which other studies not. This could increase the credibility in your future study.

Best of luck in your revision.

**References**

1. Berghuijs, W.R., Larsen, J.R., van Emmerik, T.H. and Woods, R.A., 2017. A global assessment of runoff sensitivity to changes in precipitation, potential evaporation, and other factors. *Water Resources Research*, *53*(10), pp.8475-8486.
2. Dey, P. and Mishra, A., 2017. Separating the impacts of climate change and human activities on streamflow: A review of methodologies and critical assumptions. *Journal of Hydrology*, *548*, pp.278-290.
3. Duan, L., Man, X., Kurylyk, B.L., Cai, T. and Li, Q., 2017. Distinguishing streamflow trends caused by changes in climate, forest cover, and permafrost in a large watershed in northeastern China. *Hydrological processes*, *31*(10), pp.1938-1951.
4. Jiang, C., Xiong, L., Wang, D., Liu, P., Guo, S. and Xu, C.Y., 2015. Separating the impacts of climate change and human activities on runoff using the Budyko-type equations with time-varying parameters. *Journal of Hydrology*, *522*, pp.326-338.
5. Wang, D. and Alimohammadi, N., 2012. Responses of annual runoff, evaporation, and storage change to climate variability at the watershed scale. *Water Resources Research*, *48*(5).
6. Wang, T., Yang, H., Yang, D., Qin, Y. and Wang, Y., 2018. Quantifying the streamflow response to frozen ground degradation in the source region of the Yellow River within the Budyko framework. *Journal of Hydrology*, *558*, pp.301-313.
7. Wei, X., Li, Q., Zhang, M., Giles-Hansen, K., Liu, W., Fan, H., Wang, Y., Zhou, G., Piao, S. and Liu, S., 2018. Vegetation cover—another dominant factor in determining global water resources in forested regions. *Global change biology*, *24*(2), pp.786-795.
8. Yang, D., Sun, F., Liu, Z., Cong, Z., Ni, G. and Lei, Z., 2007. Analyzing spatial and temporal variability of annual water-energy balance in nonhumid regions of China using the Budyko hypothesis. *Water Resources Research*, *43*(4).
9. Yang, H. and Yang, D., 2011. Derivation of climate elasticity of runoff to assess the effects of climate change on annual runoff. *Water Resources Research*, *47*(7).

10. Zhang, S., Yang, H., Yang, D. and Jayawardena, A.W., 2016. Quantifying the effect of vegetation change on the regional water balance within the Budyko framework. *Geophysical Research Letters*, *43*(3), pp.1140-1148.
11. Zhang, S., Yang, Y., McVicar, T.R. and Yang, D., 2018. An Analytical Solution for the Impact of Vegetation Changes on Hydrological Partitioning Within the Budyko Framework. *Water Resources Research*, *54*(1), pp.519-537.
12. Zhou, G., Wei, X., Chen, X., Zhou, P., Liu, X., Xiao, Y., Sun, G., Scott, D.F., Zhou, S., Han, L. and Su, Y., 2015. Global pattern for the effect of climate and land cover on water yield. *Nature communications*, *6*, p.5918.

---

## Referee Comment (RC3) · Anonymous Referee #3 · 22 Mar 2018

The paper by Wu et al addresses an important issue for one of the world's major river systems, namely the identification of sources of change. The paper is generally clearly written, although there are numerous errors of English that need correction. The authors correctly point out that the effects of climate change are multifaceted, and can include changes to the physical properties of the hydrological system in response to changing climate. However, since some of the earlier work that they review fails to include this point, the discussion in the paper is somewhat confusing and needs clarification. The paper presents time series of permafrost maximum frozen depth (Fig 2), which are interesting and show variable response between sites. However, the authors only point to an overall decreasing trend. This seems to be a major simplification

and the authors should consider a more detailed analysis of these data, and attempt to explain the differences, if possible, for example in terms of basin location and local climate variability. I have one important and major reservation about the extensive analysis undertaken in this paper. The authors use the Budyko relationship as a basis of their analysis of non-stationarity. The Budyko relationship is highly simplified, but extremely useful in presenting an overall global perspective on catchment behaviour – in particular, the precipitation versus energy controls on evaporation. It is however just a convenient and very approximate relationship. The authors here treat the Budyko model as correct, and use apparent differences in their parameterisation of the model as a tool to interpret change. However, inspection of Figure 5b clearly shows that, while the empirical data for the three sub-basins considered fit well within the overall envelope of response from a Budyko-type response, each has a distinct and quite different relationship, which is not captured by the overall Budyko model fit. In my opinion, this invalidates the analysis. A model of this type is so simplified that it can only be approximate and imprecise. The authors here attribute too much precision to the model and seek to identify parametric differences, which in my opinion have no physical basis – they are an artifact of the assumption of the model. I regret that in my opinion this paper is therefore not publishable. I recommend that the authors give more attention to the data as presented in Fig5b, and also consider a more physically-based modelling approach to better understand the physical controls on this system.

---

## Author Comment (AC1) · 14 Apr 2018

(a) For water storage change: The above Fig.1 is obtained from Wang et al., 2018. This figure shows the water storage change of the SRYR calculated by different methods. We draw a dark dashed line on it to indicate the zero S. As shown in this figure, S change fluctuates around the zero line but increases in 21st. These results are similarly with my results shown in first round submitted supplementary Figure 3. There is a larger deviation in 21st between recalculated w' and w caused by larger water storage change. Additionally, Figure 3 shows 11-year moving average method is efficient to remove water storage change. It is different form 5-year average value used by

[Figure]

Wang et al. 2018. And as shown in the Fig. 1, there is a deviation between water storage obtained from GRACE and the other two datasets. DS_GLEAM and DS_GBEHM show an increase of water storage change in 21st, however,DS_GRACE shows a decrease of water storage change. Wang et al., 2018 just use one landmass dataset CRS to analysis DS, but the ensemble mean (simple arithmetic mean of JPL, CSR, GFZ fields) was most effective in reducing the noise in these datasets (Sakumura et al., 2014). Different from Wang et al., 2018, the grace data we used are ensemble mean of JPL, GFZ and CRS landmass dataset. To further exam the efficiency of 11-year moving average mothed, ABCD model and Grace data are used to simulate and analyse water storage change. The results are show in supplementary 2. (b) Watershed property parameter (w) in the Fu's equation Reply: We agree that catchment properties are related to watershed slope, vegetation, soil properties and climate change. However, watershed slope is a constant value which won't change with time. In watershed underlain by frozen ground, frozen ground degradation will alter soil properties. Accordingly, our study considered climate change and frozen ground change impacts on catchment properties. A linear stepwise regression method was used to analyse potential factors impacts. And then land cover change impacts are calculated by using total discharge change (DQ) minus climatic-induced discharge change and permafrost degradation induced discharge change. Landcover change impacts are shown in Fig. 8. Following equation is used by Wang et al., 2018: Q=(P-ET_Budyko )+(-ET_dev-△S)=Q_Budyko+Q_dev (1) Where Q_budyko is long-term average of the observed annual streamflow, Q_dev is the deviation of the observed annual Q from Qbudyko. As indicated by Eq.(1), Qdev corresponds to discharge change caused by catchment properties change and neglected water storage change. Wang et al., 2018 directly attributes Q_dev to frozen ground degradation and landscape change. However, catchment properties change is closely related to relative infiltration capacity, relative soil and water storage, and average slope. For the same watershed, considering precipitation intensity, temperature and potential evaporation impacts on catchment properties is more reasonable. I think without considering climate change impacts on the catch-
ment properties are the main reason of Wang et. al., 2018 obtaining unreasonable results that permafrost degradation will decrease 70% streamflow in sub-basins above JM staion. It has been found that the permafrost degradation could enlarge baseflow in cold regions (Walvoord and Striegl, 2007; Jacques and Sauchyn, 2009; Bense et al., 2012; Evans et al., 2015; Duan et al.,2017). Decrease in MFD because of global warming was considered as a major factor for the increase in baseflow in the Qilian mountain, China (Qin et al. 2016). Additionally, the melt ice within permafrost and increasing hydrologic connectivity following permafrost degradation will increase the runoff discharge (Connon et al. 2014; Duan et al. 2017). (C) For vegetation/landcover change impacts Landcover change impacts on discharge change are analysed by Cuo et al., (2013) through VIC model. The results indicate landcover change is negligible above Tangnaihai station which is the outlet station of my study area. And in the Introduction Section, we already emphasized the low population density in the cathment above Tangnaihai hydrological station (about 6/km2, 2003 census data) and in the area above the Huangheyan station (0.34/km2) (Liang et al., 2010). From 1990 to 2000, the change in land use in the SRYR was generally less than 5% even a few of the sites exhibited 5-15% of the change (Wang et al., 2010). Accordingly, we neglected the landcover change in statistical analysis, but considered it as residual error of statistical analysis as shown in Figure 8. The results are consistent with the landcover change and low population intensity in the study area.

References Sakumura, C., Bettadpur, S., & Bruinsma, S. (2014). Ensemble prediction and intercomparison analysis of grace time‐variable gravity field models. Geophysical Research Letters, 41(5), 1389-1397. Cuo, L., Zhang, Y., Gao, Y., Hao, Z., & Cairang, L. (2013). The impacts of climate change and land cover/use transition on the hydrology in the upper Yellow River Basin, China. Journal of Hydrology, 502, 37-52. doi: 10.1016/j.jhydrol.2013.08.003 Wang, S.-Y., Liu, J.-S., & Ma, T.-B. (2010). Dynamics and changes in spatial patterns of land use in Yellow River Basin, China. Land Use Policy, 27(2), 313-323. doi: 10.1016/j.landusepol.2009.04.002 Liang, S., Ge, S., Wan, L., & Zhang, J. (2010). Can climate change cause the Yellow River to dry

up? Water Resources Research, 46(2), n/a-n/a. doi: 10.1029/2009wr007971 Wang, T., Yang, H., Yang, D., Qin, Y., & Wang, Y. (2018). Quantifying the streamflow response to frozen ground degradation in the source region of the yellow river within the budyko framework. Journal of Hydrology, 558, 301-313. Duan, L., Man, X., Kurylyk, B. L., Cai, T., & Li, Q. (2017). Distinguishing streamflow trends caused by changes in climate, forest cover, and permafrost in a large watershed in northeastern china. Hydrological Processes, 31(10). Jacques, J. M. S., & Sauchyn, D. J. (2009). Increasing winter baseflow and mean annual streamflow from possible permafrost thawing in the northwest territories, canada. Geophysical Research Letters, 36(1), 329-342. Bense, V. F., Kooi, H., Ferguson, G., & Read, T. (2012). Permafrost degradation as a control on hydrogeological regime shifts in a warming climate. Journal of Geophysical Research Earth Surface, 117(F3), 8316. Walvoord, M. A., & Striegl, R. G. (2007). Increased groundwater to stream discharge from permafrost thawing in the Yukon River basin: potential impacts on lateral export of carbon and nitrogen. Geophysical Research Letters, 34(12), 123–134. Duan, L., Man, X., Kurylyk, B. L., & Cai, T. (2017). Increasing winter baseflow in response to permafrost thaw and precipitation regime shifts in northeastern china. Water, 9(1), 1-15. Evans, S. G., Ge, S., & Liang, S. (2015). Analysis of groundwater flow in mountainous, headwater catchments with permafrost. Water Resources Research, 51(12), n/a-n/a. Qin, Y., Lei, H., Yang, D., Gao, B., Wang, Y., & Cong, Z., et al. (2016). Long-term change in the depth of seasonally frozen ground and its ecohydrological impacts in the Qilian Mountains, northeastern Tibetan Plateau. Journal of Hydrology, doi:10.1016/j.jhydrol.2016.09.008

Please also note the supplement to this comment:
https://www.hydrol-earth-syst-sci-discuss.net/hess-2017-744/hess-2017-744-AC1-supplement.pdf

[Figure]

[Figure]

**Fig. 1.** Water storage change in the SRYR obtained from Wang et al.,2018

**Supplement:**

To further exam the efficiency of 11-year moving average method in removing water storage change in study area, we use ABCD (Thomas, 1981) model to simulate the three sub-basins discharge change. Results are shown as following:

**1. Model calibration and validation**

[Figure]

Figure 1 Discharge simulated by ABCD

| Sub-basins | a | b | c | d | $G_0$ | $S_0$ | $N_c$ | $N_v$ |
| --- | --- | --- | --- | --- | --- | --- | --- | --- |
| JM | 0.96 | 500 | 0.1 | 0.1 | 100 | 100 | 0.65 | 0.63 |
| MQ | 0.98 | 550 | 0.1 | 0.1 | 260 | 100 | 0.69 | 0.64 |
| TNH | 0.96 | 400 | 0.12 | 0.1 | 200 | 50 | 0.60 | 0.55 |

The performance of ABCD model is evaluated by Nash–Sutcliffe efficiency. Generally model performance is very good if $R^2 > 0.75$, satisfactory if $0.36 < R^2 < 0.75$, and unsatisfactory if $R^2 < 0.36$ (Nashand Sutcliffe, 1970; Krause et al., 2005; Moriasi et al., 2007).

**2. Water storage variation**

[Figure]

Figure 2 Water storage change obtained by ABCD model

Here S+G indicates annual soil water and ground water storage. Blue lines indicate EWT obtained for ensemble mean of the three Grace datasets. Simulated S+G agrees well with EWT. It proves efficiency of ABCD model in simulation discharge of the three sub-basins.

**3. Δ(S+G) obtained by 11-year moving average method**

$$\Delta(S + G) = (S + G)_i - (S + G)_{i-1}$$

$(S + G)_i$ indicates each value of 11-year moving average (S+G) series shown in Fig. 2.

[Figure]

Figure 3 Water storage change

As indicated by Fig. 3, Δ(S+G) amplitudes of the three sub-basins are less than 3 mm. So by using 11-year moving average mothed, water storage change is negligible in this study area.

**Reference**

Sakumura, C., Bettadpur, S., & Bruinsma, S. (2014). Ensemble prediction and intercomparison analysis of grace time‑variable gravity field models. Geophysical Research Letters, 41(5), 1389-1397.

Thomas, H. A. (1981). Improved methods for national water assessment, Report to U.S. Water Resources Council, Contract WR15249270. Water Resour. Counc., Washington, DC: U.S. Water Resources Council.

Nash, J. E. and Sutcliffe, J. V.: River flow forecasting through conceptual models: Part I. A discussion of principles, J. Hydrol., 10,282–290, 1970.

---

## Author Comment (AC2) · 14 Apr 2018

1. In dozens of places there are grammatical errors, missing punctuation, or improper (or at least less than idea) word usage. I'd encourage the authors to have an English colleague read the paper carefully or to go through an English editing service. It is not in terrible shape, but it could be improved. Reply: This paper is further modified for reading.

2. Duan et al. (2017) tackle a similar problem to the present study but in a different approach. I'd encourage the authors to explain why their approach is superior, or at least why it might be preferred in some cases. This could occur in the intro or in

the discussion perhaps. In general, the authors should be highlighting that climate change can result in not just differences in the forcing, but also in the system itself (e.g. climate change can alter precipitation regimes which is the hydrologic loading, but it can also alter the hydrologic functioning of a landscape by altering the vegetation and permafrost or seasonally frozen ground distribution)

Duan et al. 2017. Distinguishing streamflow trends caused by changes in climate, forest cover, and permafrost in a large watershed in northeastern China. Hydrol. Process.31

Reply: Yes! We agree, this is a constructive comment. Accordingly, we discussed the modified method used in this study in Sect 5.2: In this modified analysis method, climate change directly impacts (DQc1) are calculated based on existing method and potential climate impacts (DQc2) are analysed by a linear regression analysis between catchment parameters and potential candidate variables. This mothed efficiently considered climate change impacts on catchment properties which provide a more reasonable assessment of human activity impacts. With this flexible method all potential factors can be considered in discharge change analysis by regression analysis, the residual errors of regression analysis possibly indicate impacts of the penitential factors which wasn't considered in regression analysis due to record limitation. Climate changes both impact on system and forcing are highlighted in Sect 1 as follow: Climate changes not only impact on climate forces (precipitation, temperature etc.) but also alter catchment properties (permafrost distribution, vegetation etc.) (Duan et al., 2017).

3. The introduction is generally ok, but I do not like the build up to the objectives at the end. For example, the authors state (P3, L10-13): "these relationships have not been previously examined in the SRYR". So is this just a case study then? If so, perhaps HESS would not be interested in publishing it. I would rather argue that the paper advances the science by further breaking down these offsetting or cumulative discharge disturbances, and specifically examine the role of permafrost degradation

-which has been largely ignored in past studies using these statistical approaches.

Reply: We agree. This sentence is changed into:

Permafrost degradation is mainly caused by climate change, so it should be considered in climate-induced discharge change analysis in the SRYR.

4. P5, L5, the daily frozen depth of the active layer was identified every month. This makes no sense. How do you identify something with daily frequency on a monthly basis? Also, here the authors say active layer, but is there truly an active layer everywhere - in many places there is no permafrost, correct? Anyway, this is all worded confusingly. The "frozen depth of the active layer" could mean the distance from the land surface to the bottom of the frozen zone (which unless there is a horizontal talik would mean all the way to the bottom of the permafrost during the winter) or it could mean the distance from the land surface to the top of the frozen zone, which would be better called the thawed depth. I'd suggest this be reworded Reply: We agree. Thanks for you carefully review. As you said, it is a misleading understand of active layer existed in our manuscript. Permafrost records are not available in these observed records of frozen ground. Because as shown in Figure2, all the MFD observation stations are located in the transition zone or seasonally frozen ground area. So only MFD can obtained from these stations, but the change of annual maximum frozen depth can also be used to indicate adjacent permafrost change and seasonally frozen ground change in this area. We reworded these sentences in Sect. 2.2:

Monthly mean value was obtained from the daily frozen depth of frozen ground and then used to estimate the annual maximum frozen depth (MFD) of the study period. The MFD value obtained from the monthly mean frozen depth was used to indicate the degradation of adjacent permafrost and seasonally frozen ground.

5. Figure 3 and P5, L20-25: How were these change points detected? Was any sensitivity analysis conducted on moving these change points incrementally forward or backward in time? Reply: These change points are used in a comparison anal-

СЗ

ysis between this study and previous studies. The two change points were carefully chosen in considering change points used in previous studies instead of using theorybased method to detect them. The two change points (1990, 2002) was first defined by Tang et al. (2013) according to the change of zero-flow frequency. And according to the runoff change pattern as shown in Fig. 2, by using this two change points three periods can be divided: pre-change period (1961-1990), low-flow period (1991-2002), recovery period (2003-2013). The baseline period is same as the climate pre-change period defined by IPCC (IPCC, 2007). The first change points (1990) were also adopted by Zheng et al. (2009) and Meng et al., (2016) to analyse streamflow change in the SRYR. A theory-based detect method was used by Zhao et al. (2009) and three different change points are defined: 1985,1989 and 1989 for JM, MQ and TNH, stations, respectively.

The change points adopted by Zhao et al. (2009) are different from Zheng et al. (2009) and this study, however, by using existing method we obtained similar results of catchment properties change impacts on streamflow change as indicated in Table 3 and P 11 L 29-32. 6. The transition from Eq. 7 to Equations 8-10 was hard for me to follow. This should be reworded Reply: The transition is further explained and reworded in P 7 L 21-24 as following: The sensitivity coefficients , and in Eq. 7 are defined as partial derivative to respective variables. To eliminate discretization errors, the mean value of the sensitivity coefficients in the pre-change and post-change periods: , and are defined and used in following study(Jiang et al., 2015): 7. P8, L14-17, a key message of this paper is that climate change can influence the catchment specific parameter. This is likely true, but it should be explained rather than just citing a few other studies. If this is so important, the authors should provide some examples of why this might be from a physical perspective. Reply: P8. L14-17 revised as following: However, this is not true because catchment relative infiltration capacity and soil water storage are also related to climate factors: precipitation intensity and potential evapotranspiration, respectively. (Yang et al., 2007;), the catchment specific parameter can also be influenced by climate type (Williams et al., 2012) and climate change (Jiang et al. 2015).

8. Related to the above, the paper would be much improved by the authors tying in the physical hydrologic environment to the statistical analysis results. For example, why would rainfall-runoff processes be altered by changes in permafrost, particularly in the source region of the Yellow River. What soil is there? What slope? Based on this why might the runoff ratios change? Without tying the results to the physical setting, the entire results section comes across as a bit of an arm waving exercise Reply: Accept, according to this comment, Table 2 is added in the modified manuscript. Indeed! This article lack of physical explanations due to these analyses are based on statistical method. Potential explanations are added in Sect. 5.2 However, potential physical explanations can be found in previous studies. It has been found that the permafrost degradation could enlarge baseflow in cold regions (Walvoord and Striegl, 2007; Jacques and Sauchyn, 2009; Bense et al., 2012; Evans et al., 2015; Duan et al.,2017). Decrease in MFD because of global warming was considered as a major factor for the increase in baseflow in the Qilian mountain, China (Qin et al. 2016). Additionally, the melt ice within permafrost and increasing hydrologic connectivity fallowing permafrost thaw-induced land-cover change will increase the runoff discharge (Connon et al. 2014; Duan et al. 2017). According to these research, Further deeper investigations are required to link the rainfall-runoff and base flow behaviors with the physical mechanism of frozen soils and performed in the SRYR.

Minor comments I'd invite the authors to refer to Wang et al. 2018, which has some overlap with the present study (especially geographically) along themes related to changing MFD. They may be able to feed some data from this prior paper into their statistical approach.

Wang et al. 2018. Historical and future changes of frozen ground in the upper Yellow River Basin. Global and Planetary Change, 172, 199-2011

P1, L29 'are facing serious water shortages' – use of the present tense here might warrant a more recent citation as Yang et al. (2004) is now 14 years old. Reply: Modified as follow: Due to the dry climate and heavy water demands, people in the

Yellow River basin are facing serious water shortages in 1990s.

P3, L7-8, 'permafrost thawing to surface water discharge' - I don't really like this wording. It seems to imply that permafrost thaw produces streamflow (i.e. the meltwater is a significant contributor to streamflow). If so, that would surely be an incredibly fast thaw rate! Reply: Accept. Modified as follow: Permafrost degradation will increase the depth and length of subsurface flow paths and the lag-times of subsurface water flow from infiltration to surface water discharge.

P3, L30-40, I find this description confusing. The opposite ends of the permafrost spectrum are continuous and isolated, so why would the authors lump those into a single 'continuous' category. Also Figure 1 shows alpine permafrost, but does not indicate if this is continuous or discontinuous. I guess that is what the authors are explaining in that paragraph – the classification system is not standard Reply: Accept! The classification of permafrost was further explained in Sect. 2.1 P4 L1-7. For this map, there are three permafrost classifications: predominantly continuous permafrost (70-80%), isolated permafrost (40-60%) and alpine permafrost. This classification scheme is different from that of the International Permafrost Association (IPA) (Cheng and Wu, 2007; Ren et al., 2012). In Figure 1b, the predominantly continuous permafrost and the isolated permafrost are further combined into the plateau permafrost in the Tibetan Plateau (Ren et al., 2012). Figure 1 was further modified according to this comment and attached in this reply. P4, L3, Why is the SRYR unique? This is not explained Reply: It is explained in P4 L 6-7. Due to the water resource significance and unique landscape, the SRYR provides an ideal location to observe the hydrological effects of degrading permafrost with climate change. P17, L15, what about aerial geophysical methods in permafrost? See Minsley et al. Minsley et al. 2012, Airborne electromagnetic imaging of discontinuous permafrost. Geophys. Res. Lett. Reply: Yes! Indeed! Airborne electromagnetic method is different from those classical methods. This kind of method can be employed in a larger area catchment. However, long-term dynamic state of permafrost is difficult to be obtained by this kind of method.

P17, L18-24, this paragraph is worded as though it were a key springboard to future work. I found the logic in the section hard to follow. Perhaps the wrong word is used in some sentence, or perhaps my mind is dense after another long day. But they seem to suggest that decreasing MFD is a positive factor and that this is enigmatic – but then they indicate later that Qin et al. 2016 showed this. Are they saying there are no physical explanation for this or that it is unusual? Reply: Qin et al. 2016 did show that positive correlation between decreased MFD and increased baseflow. However, it just emphasises permafrost degradation impacts on groundwater discharge are different from total discharge analysed in this study. Potential physical explanations were added in Section 5.2 as mentioned in reply of comments 8.

**Fig. 1.** Figure 1 (a) Location map of the source region of the Yellow River (JM, MQ and TNH) (b) Distribution of permafrost, meteorological and hydrological stations (The red circles indicates stations which h

---

## Author Comment (AC3) · 15 Apr 2018

(1) In Fig 5b it appears that the classic Budyko curve does not capture the local relationship type-curves well. This is because in our study we use a time varying average to calculate time-varying points. And therefore, each point has a different w value for each time period which is observed in Fig 5b as the distribution of points for each sub-basin. If we were to draw many more type-curves for different w values, all our points will be located on a classical Budyko type curve. Fig. 5b just show three curves with more than 120 time-varying points to help readers see the range of w values. Further, the classic Budyko method is obviously efficient for this study, because all the w values

are calculated from Fu's equation which plotted in Fig. 5b as attached in following. As such we have further clarified the text. Specifically, in section 4.2 we added: Points in the Budyko space can be located respective to a given Budyko-type curve by changing the w value for Fu's equation. Example classical Budyko curves for w = 2.4,2.0 and 1.7 are shown in Fig. 5(b). The gray area of Fig. 5b indicates where the Budyko type-curve changes with different w values. (2) MFD variations have been discussed in Sect. 2.2 The mean frozen depth exhibits a decreasing trend at most of the observation sites, especially in the period after 1980. And 11-year moving average value of the selected for the further analysis. We did not describe these variations clearly, however, all the analyses are based on realistic data. It won't impact the analysis results. (3) The reliability of this research has been discussed in Sect. 5.2: Considering permafrost degradation in a whole catchment hydrological system is more complexed, but it is necessary for better understanding and predicting earth system and hydrological change in cold regions. At present, there is few physical models can accurately handle these situations. But it can be preliminarily considered by using a conceptual model and statistic methods. And interesting results obtained by considering permafrost at a catchment hydrological scale, degrading permafrost that can role as a positive factor for discharge change. And the potential physical explanations are further discussed in Sect. 5.2: It has been found that the permafrost degradation could enlarge baseflow in cold regions (Walvoord and Striegl, 2007; Jacques and Sauchyn, 2009; Bense et al., 2012; Evans et al., 2015; Duan et al.,2017). Decrease in MFD because of global warming was considered as a major factor for the increase in baseflow in the Qilian mountain, China (Qin et al. 2016). Additionally, the melt ice within permafrost and increasing hydrologic connectivity fallowing permafrost thaw-induced land-cover change will increase the runoff discharge (Connon et al. 2014; Duan et al. 2017).

References

Connon, R. F., Quinton, W. L., Craig, J. R., & Hayashi, M. (2014). Changing hydrologic connectivity due to permafrost thaw in the lower liard river valley, nwt, canada. Hydrological Processes, 28(14), 4163-4178. Duan, L., Man, X., Kurylyk, B. L., Cai, T., & Li, Q. (2017). Distinguishing streamflow trends caused by changes in climate, forest cover, and permafrost in a large watershed in northeastern china. Hydrological Processes, 31(10). Jacques, J. M. S., & Sauchyn, D. J. (2009). Increasing winter baseflow and mean annual streamflow from possible permafrost thawing in the northwest territories, canada. Geophysical Research Letters, 36(1), 329-342. Bense, V. F., Kooi, H., Ferguson, G., & Read, T. (2012). Permafrost degradation as a control on hydrogeological regime shifts in a warming climate. Journal of Geophysical Research Earth Surface, 117(F3), 8316. Walvoord, M. A., & Striegl, R. G. (2007). Increased groundwater to stream discharge from permafrost thawing in the Yukon River basin: potential impacts on lateral export of carbon and nitrogen. Geophysical Research Letters, 34(12), 123–134. Duan, L., Man, X., Kurylyk, B. L., & Cai, T. (2017). Increasing winter baseflow in response to permafrost thaw and precipitation regime shifts in northeastern china. Water, 9(1), 1-15. Evans, S. G., Ge, S., & Liang, S. (2015). Analysis of groundwater flow in mountainous, headwater catchments with permafrost. Water Resources Research, 51(12), n/a-n/a. Qin, Y., Lei, H., Yang, D., Gao, B., Wang, Y., & Cong, Z., et al. (2016). Long-term change in the depth of seasonally frozen ground and its ecohydrological impacts in the Qilian Mountains, northeastern Tibetan Plateau. Journal of Hydrology, doi:10.1016/j.jhydrol.2016.09.008

[Figure]

**Fig. 1.** Figure5b

---

## Author Comment (AC4) · 22 Apr 2018

**1. In dozens of places there are grammatical errors, missing punctuation, or improper (or at least less than idea) word usage. I'd encourage the authors to have an English colleague read the paper carefully or to go through an English editing service. It is not in terrible shape, but it could be improved.**

**Reply:**

This paper is further modified for reading.

**2. Duan et al. (2017) tackle a similar problem to the present study but in a different approach. I'd encourage the authors to explain why their approach is superior, or at least why it might be preferred in some cases. This could occur in the intro or in the discussion perhaps. In general, the authors should be highlighting that climate change can result in not just differences in the forcing, but also in the system itself (e.g. climate change can alter precipitation regimes which is the hydrologic loading, but it can also alter the hydrologic functioning of a landscape by altering the vegetation and permafrost or seasonally frozen ground distribution)**

**Duan et al. 2017. Distinguishing streamflow trends caused by changes in climate, forest cover, and permafrost in a large watershed in northeastern China. Hydrol. Process.31**

**Reply:**

Yes! We agree, this is a constructive comment. Accordingly, we discussed the modified method used in this study in Sect 5.2:

Through this modified analysis method, climate change directly impacts ($\Delta Q_{c1}$) are calculated based on existing method, potential climate impacts ($\Delta Q_{c2}$) are analysed by a stepwise linear regression analysis between catchment parameters and potential candidate variables. This method efficiently considered climate change impacts on catchment properties which provide a more reasonable assessment of human activity impacts. With this flexible method all potential factors can be considered in discharge change analysis by stepwise regression analysis, the residual errors of regression analysis possibly indicate impacts of the penitential factors which wasn't directly considered in regression analysis due to record limitation.

Climate changes both impact on system characteristics and system forcing are highlighted in Sect 1 as follow:

Climate changes not only impact on climate forces (precipitation, temperature etc.) but also alter catchment properties (permafrost distribution, vegetation etc.) (Duan et al., 2017).

**3. The introduction is generally ok, but I do not like the build up to the objectives at the end. For example, the authors state (P3, L10-13): "these relationships have not been previously examined in the SRYR". So is this just a case study then? If so, perhaps HESS would not be interested in publishing it. I would rather argue that the paper advances the science by further breaking down these offsetting or cumulative discharge disturbances, and specifically examine the role of permafrost degradation –which has been largely ignored in past studies using these statistical approaches.**

**Reply:**

We agree. This sentence is changed into:

Permafrost degradation is mainly caused by climate change therefore it should be considered in climate-induced discharge change analysis in the SRYR.

**4. P5, L5, the daily frozen depth of the active layer was identified every month. This makes no sense. How do you identify something with daily frequency on a monthly basis? Also, here the authors say active layer, but is there truly an active layer everywhere – in many places there is no permafrost, correct? Anyway, this is all worded confusingly. The "frozen depth of the active layer" could mean the distance from the land surface to the bottom of the frozen zone (which unless there is a horizontal talik would mean all the way to the bottom of the permafrost during the winter) or it could mean the distance from the land surface to the top of the frozen zone, which would be better called the thawed depth. I'd suggest this be reworded**

**Reply:**

We agree. Thanks for your carefully review. As you said, there is a misleading understand of active layer existed in our manuscript. Permafrost records are not available in these observed records of frozen ground. Because as shown in Figure 2, all the MFD observation stations are located in the transition zone or seasonally frozen ground area. So only MFD can obtained from these stations, but the change of annual maximum frozen depth can also be used to indicate adjacent permafrost change and seasonally frozen ground change in this area. We reworded these sentences in Sect. 2.2:

Monthly mean value was obtained from the daily frozen depth of frozen ground and then used to estimate the annual maximum frozen depth (MFD) of the study period. The MFD value obtained from

**5. Figure 3 and P5, L20-25: How were these change points detected? Was any sensitivity analysis conducted on moving these change points incrementally forward or backward in time?**

**Reply:**

These change points are used in a comparison analysis between this study and previous studies. The two change points were carefully chosen in considering change points used in previous studies instead of using theory-based method to detect them.

The two change points (1990, 2002) was first defined by Tang et al. (2013) according to the change of zero-flow frequency. And according to the runoff change pattern as shown in Fig. 2, by using this two change points three periods can be divided: pre-change period (1961-1990), low-flow period (1991-2002), recovery period (2003-2013). The baseline period is same as the climate pre-change period defined by IPCC (IPCC, 2007). The first change points (1990) were also adopted by Zheng et al. (2009) and Meng et al., (2016) to analyse streamflow change in the SRYR. A theory-based detect method was used by Zhao et al. (2009) and three different change points are defined: 1985,1989 and 1989 for JM, MQ and TNH, stations, respectively.

The change points adopted by Zhao et al. (2009) are different from Zheng et al. (2009) and this study, however, by using existing method we obtained similar results of catchment properties change impacts on streamflow change as indicated in Table 4 and P 11 L 29-32.

**6. The transition from Eq. 7 to Equations 8-10 was hard for me to follow. This should be reworded**

**Reply:**

The transition is further explained and reworded in P 7 L 21-24 as following:

The sensitivity coefficients $f_P'$, $f_{E_0}'$ and $f_w'$ in Eq. 7 are defined as partial derivative to respective variables. To eliminate discretization errors, the mean value of the sensitivity coefficients in the pre-change and post-change periods: $f_P^*$, $f_{E_0}^*$ and $f_w^*$ are defined and used in following study(Jiang et al., 2015):

**7. P8, L14-17, a key message of this paper is that climate change can influence the catchment specific parameter. This is likely true, but it should be explained rather than just citing a few other studies. If this is so important, the authors should provide some examples of why this might be from a physical perspective.**

**Reply:**

P8. L14-17 revised as following:

However, this is not true because catchment relative infiltration capacity and relative soil water storage are also related to climate factors: precipitation intensity and potential evapotranspiration, respectively. (Yang et al., 2007;), the catchment specific parameter can also be influenced by climate variability (Donohue et al., 2012), climate type (Williams et al., 2012) and climate change (Jiang et al. 2015).

**8. Related to the above, the paper would be much improved by the authors tying in the physical hydrologic environment to the statistical analysis results. For example, why would rainfall-runoff processes be altered by changes in permafrost, particularly in the source region of the Yellow River. What soil is there? What slope? Based on this why might the runoff ratios change? Without tying the results to the physical setting, the entire results section comes across as a bit of an arm waving exercise**

**Reply:**

Accept, according to this comment, Table 2 is added in the modified manuscript.

Indeed! This article lack of physical explanations due to these analyses are based on statistical method. Potential explanations are added in Sect. 5.2

However, potential physical explanations can be found in previous studies. It has been found that the permafrost degradation could enlarge baseflow in cold regions (Walvoord and Striegl, 2007; Jacques and Sauchyn, 2009; Bense et al., 2012; Evans et al., 2015; Duan et al.,2017). Decrease in MFD because of global warming was considered as a major factor for the increase in baseflow in the Qilian mountain, China (Qin et al. 2016). Additionally, the melt ice within permafrost and increasing hydrologic connectivity fallowing permafrost thaw-induced land-cover change will increase the runoff discharge (Connon et al. 2014; Duan et al. 2017). According to these research, further deeper investigations are required to link the rainfall-runoff and base flow behaviors with the physical mechanism of frozen soils and performed in the SRYR.

**Minor comments**

**I'd invite the authors to refer to Wang et al. 2018, which has some overlap with the present study (especially geographically) along themes related to changing MFD. They may be able to feed some data from this prior paper into their statistical approach.**

**Wang et al. 2018. Historical and future changes of frozen ground in the upper Yellow River Basin. Global and Planetary Change, 172, 199-2011**

Accept! Cited in Sect 1, There is about 23% of the permafrost in the SRYR degraded to seasonally

10  frozen ground during the past 5 decades (Wang et al., 2018).

**P1, L29 'are facing serious water shortages' – use of the present tense here might warrant a more recent citation as Yang et al. (2004) is now 14 years old.**

Reply:

15  Modified as follow:

Due to the dry climate and heavy water demands, people in the Yellow River basin are facing serious water shortages in 1990s.

**P3, L7-8, 'permafrost thawing to surface water discharge' - I don't really like this wording.**

20  **It seems to imply that permafrost thaw produces streamflow (i.e. the meltwater is a significant contributor to streamflow). If so, that would surely be an incredibly fast thaw rate!**

Reply:

Accept. Modified as follow:

25  Permafrost degradation will increase the depth and length of subsurface flow paths and the lag-times of subsurface water flow from infiltration to surface water discharge.

**P3, L30-40, I find this description confusing. The opposite ends of the permafrost**

**spectrum are continuous and isolated, so why would the authors lump those into a**

**single 'continuous' category. Also Figure 1 shows alpine permafrost, but does not**

**indicate if this is continuous or discontinuous. I guess that is what the authors are**

**explaining in that paragraph – the classification system is not standard**

5 **Reply:**

Accept! The classification of permafrost was further explained in Sect. 2.1 P4 L1- 7.

For this map, there are three permafrost classifications: predominantly continuous permafrost (70-80%), isolated permafrost (40-60%) and alpine permafrost. This classification scheme is different from that of the International Permafrost Association (IPA) (Cheng and Wu, 2007; Ren et al., 2012). In Figure 1b,

10 the predominantly continuous permafrost and the isolated permafrost are further combined into the plateau permafrost in the Tibetan Plateau (Ren et al., 2012).

Figure 1 was further modified according to this comment and attached in this reply.

**P4, L3, Why is the SRYR unique? This is not explained**

**Reply:**

15 It is explained in P4 L 6-7. Due to the water resource significance and unique landscape, the SRYR provides an ideal location to observe the hydrological effects of degrading permafrost with climate change.

**P17, L15, what about aerial geophysical methods in permafrost? See Minsley et al.**

**Minsley et al. 2012, Airborne electromagnetic imaging of discontinuous permafrost.**

20 **Geophys. Res. Lett.**

**Reply:**

Yes! Indeed! Airborne electromagnetic method is different from those classical methods. This kind of method can be employed in a larger area catchment. However, long-term dynamic state of permafrost is difficult to be obtained by this kind of method.

**P17, L18-24, this paragraph is worded as though it were a key springboard to future**

**work. I found the logic in the section hard to follow. Perhaps the wrong word is used**

**in some sentence, or perhaps my mind is dense after another long day. But they seem**

**to suggest that decreasing MFD is a positive factor and that this is enigmatic – but**

30 **then they indicate later that Qin et al. 2016 showed this. Are they saying there are no**

**physical explanation for this or that it is unusual?**

**Reply:**

Qin et al. 2016 did show that positive correlation between decreased MFD and increased baseflow. However, it just emphasises permafrost degradation impacts on groundwater discharge are different

5 from total discharge analysed in this study. Potential physical explanations were added in Section 5.2 as mentioned in reply of comments 8.

[revised manuscript text omitted]
15  (permafrost distribution, vegetation) (Duan et al., 2017). Permafrost degradation will increase the depth and length of subsurface flow paths and the lag-times of subsurface water flow from infiltration to surface water discharge (Frampton et al., 2011, Kurylyk et al., 2014). Degrading permafrost could also accelerate the flow recession process and increase local hydrological circulation (Lyon and Destouni, 2010; Lyon et al., 2009; Cuo et al., 2015), change aquifer permeability, and increased base flow
20  (Walvoord and Striegl, 2007; Bense et al., 2012; Evans et al. 2015). Permafrost degradation is mainly caused by climate change therefore it should be considered in climate-induced discharge change analysis in the SRYR.

[revised manuscript text omitted]

We do not correct the meteorological data for elevation effects because the properties of the elevation
5    effects are poorly known in the SRYR. However, it is expected that the elevation-dependent distributions of these factors are mostly included within the IDW results. In general, the elevation of ground surface in the study area increases gradually from the east to the west and creates a climate gradient parallel to the altitude gradient that is exhibited in the differences in the observed meteorological data (Figure 1b). For the annual precipitation, we found that these IDW data at the
10   catchment scale agreed with TRMM data (Tropical Rainfall Measurement Mission) 3B42 data (Tong et al., 2014) which show a great performance in streamflow simulation in upper Yellow and Yangtze River basins on the Tibetan Plateau (Hao et al., 2014). The IDW interpolation is adopted in this study to fully capture the collected $P$, $E_0$, $T$ and $I$ data at the weather stations.

15   Daily observations of frozen ground at 11 meteorological stations (Figure 1b) was collected by CMA. Monthly mean value was obtained from the daily frozen depth of frozen ground and then used to estimate annual maximum frozen depth (MFD) of the study period. The MFD value obtained from the monthly mean frozen depth was used to indicate the degradation of adjacent permafrost and seasonally frozen ground. Typical variation patterns of the mean frozen depth are shown in Figure 2. The mean
20   frozen depth exhibits a decreasing trend at most of the observation sites, especially in the period after 1980. The MFD data of the observation stations in and around each sub-basin show the same tend (Figure 2). In considering of the different record lengths of MFD in different observation stations, a station with the longest record is selected to account for the representative MFD data in each sub-basin. Therefore, M5, M10, and M14 are selected for JM, MQ and TNH, respectively, as that shown in Figure
25   1b. The station M14 is not located in the MQ sub-basin but it exhibits the same variation trend of MFD with that observed at stations in the MQ basin (M12, M13 and M16) with shorter records. However, M17 shows little variation compared to the other sites because it is located at alpine permafrost and therefore is not used in this study.

30   Figure 3 shows that discharge change can be divided into three periods: pre-change period from 1961-1990, low-flow period from 1991-2002 and recent period from 2003-2013. The two change points (1990, 2002) was first defined by Tang et al. (2013) according to the change of zero-flow frequency. The baseline period is same as the climate pre-change period defined by IPCC (IPCC, 2007).

**3 Methods**

[revised manuscript text omitted]

$$\Delta Q = f_P'\Delta P + f_{E_0}'\Delta E_0 + f_w'\Delta w \qquad (7)$$

The sensitivity coefficients $f_P'$, $f_{E_0}'$ and $f_w'$ in Eq. 7 are defined as partial derivative to respective variables ($P$, $E_0$ and $w$). To eliminate discretization errors, the mean value of the sensitivity coefficients

in the pre-change and post-change periods: $f_P^*$ , $f_{E_0}^*$ and $f_w^*$ are defined and used in following study(Jiang et al., 2015):

$$f_P^* = \frac{1}{2}\left[ f_P'\left(P, E_0, w\right) + f_P'\left(P + \Delta P, E_0 + \Delta E_0, w + \Delta w\right) \right] \tag{8}$$

$$f_{E_0}^* = \frac{1}{2}\left[ f_{E_0}'\left(P, E_0, w\right) + f_{E_0}'\left(P + \Delta P, E_0 + \Delta E_0, w + \Delta w\right) \right] \tag{9}$$

$$f_w^* = \frac{1}{2}\left[ f_w'\left(P, E_0, w\right) + f_w'\left(P + \Delta P, E_0 + \Delta E_0, w + \Delta w\right) \right] \tag{10}$$

The climatic and anthropic changes in discharge are considered as

$$\Delta Q_c = f_P^* \Delta P + f_{E_0}^* \Delta E_0, \ \Delta Q_h = \Delta Q_w = f_w^* \Delta w \tag{11}$$

where $\Delta Q_w$ is applied in this study to denote the change in discharge induced by the change in the specific catchment parameter. This method estimates the variable $w$ values and assumes that $\Delta w$ is

10 caused by the human activities such as the change in land use.

**3.2.3 Decomposition method**

For the decomposition method (Wang and Hejazi, 2011), different parts of the change in discharge are estimated from:

$$\Delta Q_c = f\left(P + \Delta P, E_0 + \Delta E_0, w\right) - f\left(P, E_0, w\right) \tag{12}$$

$$\Delta Q_h = f\left(P, E_0, w + \Delta w\right) - f\left(P, E_0, w\right) = \Delta Q - \Delta Q_c \tag{13}$$

From Eq. 13 we see that this method also attributes $\Delta w$ to human activities.

**3.3 Identifying the climatic impact on the catchment specific parameter**

20 In the existing approaches, the change in the catchment specific parameter and the induced change in the discharge are attributed to human activities, *e.g.*, change in land use. However, this is not true because catchment relative infiltration capacity and relative soil water storage are also related to climate factors: precipitation intensity and potential evapotranspiration, respectively (Yang et al., 2007;), 
[revised manuscript text omitted]

15  potential climate impacts ($\Delta Q_{c2}$) are analysed by a stepwise linear regression analysis between catchment parameters and potential candidate variables. This method efficiently considered climate change impacts on catchment properties which provide a more reasonable assessment of human activity impacts. With this flexible method all potential factors can be considered in discharge change analysis by stepwise regression analysis, the residual errors of regression analysis possibly indicate

20  impacts of the penitential factors which wasn't directly considered in regression analysis due to record limitation. An important finding from our research is that by considering permafrost at the catchment hydrological scale, degrading permafrost is a positive factor for discharge change.

[revised manuscript text omitted]

Donohue, R. J., Roderick, M. L., & Mcvicar, T. R. (2012). Roots, storms and soil pores: incorporating key ecohydrological processes into budyko's hydrological model. Journal of Hydrology, 436-437, 35-50.

Duan, L., Man, X., Kurylyk, B. L., & Cai, T. (2017). Increasing winter baseflow in response to permafrost thaw and precipitation regime shifts in northeastern china. Water, 9(1), 1-15.

Duan, L., Man, X., Kurylyk, B. L., Cai, T., & Li, Q. (2017). Distinguishing streamflow trends caused by changes in climate, forest cover, and permafrost in a large watershed in northeastern china. Hydrological Processes, 31(10).

Frampton, A., Painter, S., Lyon, S. W., & Destouni, G. (2011). Non-isothermal, three-phase simulations of near-surface flows in a model permafrost system under seasonal variability and climate change. Journal of Hydrology, 403(3-4), 352-359. doi: 10.1016/j.jhydrol.2011.04.010

Fu, B. P. (1981), On the calculation of the evaporation from land surface (in Chinese), Sci. Atmos. Sin., 5, 23–31.

Goodrich, L. E. (1982). The influence of snow cover on the ground thermal regime. Canadian Geotechnical Journal, 19(4), 421-432.

Hair J.F.J., Anderson R.E., Tatham R.L., & Black W.C. (1995) Multivariate Data Analysis. New York: Macmillan.

Hao, Z. C., Tong, K., Liu, X. L., & Zhang, L. L. (2014). Capability of tmpa products to simulate streamflow in Upper Yellow and Yangtze River basins on Tibetan Plateau. Water Science and Engineering, 7(3), 237-249.

Hu, Y., Maskey, S., Uhlenbrook, S., & Zhao, H. (2011). Streamflow trends and climate linkages in the source region of the Yellow River, China. Hydrological Processes, 25(22), 3399-3411. doi: 10.1002/hyp.8069

Immerzeel, W. W., van Beek, L. P., & Bierkens, M. F. (2010). Climate change will affect the Asian water towers. Science, 328(5984), 1382-1385. doi: 10.1126/science.1183188

IPCC (2007). Climate change 2007: The Physical Science Basis. In: Contribution of Working Group I to the Fourth Assessment Report of the Intergovernmental Panel on Climate Change, Cambridge University Press, Cambridge, 996 pp.

Jacques, J. M. S., & Sauchyn, D. J. (2009). Increasing winter baseflow and mean annual streamflow from possible permafrost thawing in the northwest territories, canada. Geophysical Research Letters, 36(1), 329-342.

[revised manuscript text omitted]

Table 2 Sub-basins characteristics

|  | Slop(°) | Area(km$^2$) | Permafrost (%) | Seasonally Frozen Ground (%) |
|---|---|---|---|---|
| JM | 5.86 | 45308 | 80% | 20% |
| MQ | 11.21 | 40920 | 23% | 77% |
| TNH | 14.03 | 36695 | 46% | 54% |

Table 3 Proposed formulas of $E/P$ versus $E_0/P$ in the Budyko framework

| Types | Solutions | Equations |
|---|---|---|
| Non-parametric formulas | Schreiber (Schreiber,1901) | $1-e^{\frac{E_0}{P}}$ |
| | Ol'dekop (Ol'dekop,1911) | $\frac{E_0}{P}\tanh\left(\frac{P}{E_0}\right)$ |
| | Budyko (Budyko, 1974) | $\left[\frac{E_0}{P}\tanh\left(\frac{P}{E_0}\right)\left(1-e^{-\frac{E_0}{P}}\right)\right]^{\frac{1}{2}}$ |
| One-parameter formulas | Fu (Fu,1981;Zhang et al., 2004) | $1+\frac{E_0}{P}-\left(1+\left(\frac{E_0}{P}\right)^{w}\right)^{\frac{1}{w}}$ |
| | Choudhury (Choudhury,1999; Yang et al., 2008) | $\left(1+\left(\frac{E_0}{P}\right)^{-n}\right)^{-\frac{1}{n}}$ |
| | Zhang (Zhang et al., 2001) | $\dfrac{1+\alpha\cdot\dfrac{E_0}{P}}{1+\alpha\cdot\dfrac{E_0}{P}+\dfrac{P}{E_0}}$ |

Table 4 Results of partitioned changes in discharge using existing methods

| Sub-basins | Periods | $w$ | | | $\Delta Q_c$ (%) | | | $\Delta Q_h$ ($\Delta Q_w$) (%) | | | $\Delta Q$ (%) |
|---|---|---|---|---|---|---|---|---|---|---|---|
| | | C* | S* | D* | C | S | D | C | S | D | |
| JM | 1961-1990 | 2 | 2.10 | 2.10 | | | Reference period | | | | |
| | 1991-2002 | 2 | 2.33 | 2.33 | -8.10 | -7.25 | -7.69 | -19.62 | -20.67 | -20.03 | -27.72 |
| | 2003-2013 | 2 | 2.21 | 2.21 | 19.57 | 16.37 | 18.60 | -13.35 | -10.05 | -12.38 | 6.22 |
| MQ | 1961-1990 | 2 | 1.85 | 1.85 | | | Reference period | | | | |
| | 1991-2002 | 2 | 2.01 | 2.01 | -11.04 | -11.20 | -11.43 | -10.85 | -10.74 | -10.47 | -21.89 |
| | 2003-2013 | 2 | 2.08 | 2.08 | -1.29 | -1.35 | -1.18 | -15.31 | -13.52 | -15.42 | -16.60 |
| TNH | 1961-1990 | 2 | 1.74 | 1.74 | | | Reference period | | | | |
| | 1991-2002 | 2 | 1.94 | 1.94 | -8.20 | -8.89 | -9.37 | -19.75 | -19.25 | -18.58 | -27.94 |
| | 2003-2013 | 2 | 1.86 | 1.86 | 6.84 | 7.34 | 8.38 | -11.86 | -11.05 | -13.40 | -5.02 |

\* The C, S and D indicate the climate elasticity method, sensitivity method and decomposition method, respectively.

Table 5 Results of $w$ covariate analysis for the period between 1961 and 2013

| Sub-basins | Models of Eq. (15) (Candidates $\overline{P}$, $\overline{E_0}$ and $\overline{T}$) | $R^2$ | Models of Eq. (15) (Candidates $\overline{I}$, $\overline{E_0}$ and $\overline{T}$) | $R^2$ |
|---|---|---|---|---|
| JM | $4.254+0.357\overline{T}-1.715\overline{P}$ | 0.67 | $5.229+0.466\overline{T}-2.582\overline{I}$ | 0.83 |
| MQ | $-1.743-1.201\overline{P}+4.874\overline{E_0}$ | 0.85 | $-3.778-1.582\overline{I}+7.291\overline{E_0}$ | 0.88 |
| TNH | $3.814+0.287\overline{T}-2.281\overline{P}$ | 0.90 | $1.982+0.442\overline{T}-3.194\overline{I}+2.591\overline{E_0}$ | 0.91 |

Table 6 MFD record length and correlation between CMAT and MAT at observation stations

| Station | $R^2$ of MFD- CMAT | $R^2$ of MFD- MAT | MFD record length (years) |
|---------|--------------------|--------------------|----------------------------|
| M5 | 0.65 (–)* | 0.49 (–) | 1960-2004(45) |
| M6 | 0.29 (–) | 0.13 (–) | 1983-2004(22) |
| M7 | 0.34 (–) | 0.053 (–) | 1980-2004(25) |
| M8 | 0.18 (–) | 0.41 (–) | 1990-2004(13) |
| M10 | 0.21 (–) | 0.08 (–) | 1962-2004(43) |
| M11 | 0.01 (–) | 0.18 (–) | 1991-2004(14) |
| M12 | 0.55 (–) | 0.46 (–) | 1967-1997(31) |
| M13 | 0.58 (–) | 0.41 (–) | 1974-2004(31) |
| M14 | 0.47 (–) | 0.24 (–) | 1966-2004(39) |
| M16 | 0.47 (–) | 0.38 (–) | 1971-1979  1995-2004(28) |
| M17 | 0.52 (–) | 0.35 (–) | 1974-2004(31) |

5 *(–) indicates negative correlation